# INTERMEDIATE LAYER CLASSIFIERS FOR OOD GENERALIZATION

**Arnas Uselis**
Tübingen AI Center
University of Tübingen
`arnas.uselis@uni-tuebingen.de`

**Seong Joon Oh**
Tübingen AI Center
University of Tübingen

## ABSTRACT

Deep classifiers are known to be sensitive to data distribution shifts, primarily due to their reliance on spurious correlations in training data. It has been suggested that these classifiers can still find useful features in the network's last layer that hold up under such shifts. In this work, we question the use of last-layer representations for out-of-distribution (OOD) generalisation and explore the utility of intermediate layers. To this end, we introduce *Intermediate Layer Classifiers* (ILCs). We discover that intermediate layer representations frequently offer substantially better generalisation than those from the penultimate layer. In many cases, zero-shot OOD generalisation using earlier-layer representations approaches the few-shot performance of retraining on penultimate layer representations. This is confirmed across multiple datasets, architectures, and types of distribution shifts. Our analysis suggests that intermediate layers are less sensitive to distribution shifts compared to the penultimate layer. These findings highlight the importance of understanding how information is distributed across network layers and its role in OOD generalisation, while also pointing to the limits of penultimate layer representation utility. Code is available at `https://github.com/oshapio/intermediate-layer-generalization`.

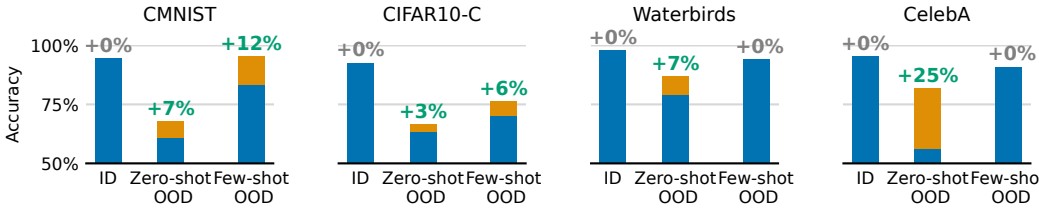

Figure 1: **Using last vs intermediate layers for OOD generalisation**. A common way to address distribution shift is to fine-tune the last layer of a network on the target distribution (few-shot learning). We show that earlier-layer representations often generalise better than the last layer. Moreover, even when only the in-distribution (ID) data is available, earlier-layer representations are often better than the last layer (zero-shot learning).

## 1 INTRODUCTION

Deep neural networks (DNNs) often lack robustness when evaluated on out-of-distribution (OOD) samples: once a classifier is trained, its performance often drops significantly when deployed in a new environment (Taori et al., 2020). A reason for this is that training data often contain spurious correlations (Singla & Feizi, 2021; Li et al., 2023), which encourage models to learn shortcuts (Scimeca et al., 2022; Cadene et al., 2020; Recht et al., 2019). Relying on such shortcuts leads to models that do not generalize well to out-of-distribution (OOD) samples (Geirhos et al., 2020; 2022; Rosenfeld et al., 2018; Beery et al., 2018) lying outside the training distribution. Many attempts have been made address the disparity in performance between in-distribution (ID) and OOD samples (Zhang et al., 2018; Yun et al., 2019; Shi et al., 2022; Verma et al., 2019), but when no testing data is available from the target distribution, generalization is challenging.

Recent work has shown that the last layer of DNNs already contain enough information for generalization in the cases of long-tail classification (Kang et al., 2020), domain generalization (Rosenfeld et al., 2022), and learning under spurious correlations (Kirichenko et al., 2023). In these works, only the linear classifier of the last layer is retrained on the target distribution, and the model is shown to generalize well to OOD samples. This suggests that the model can learn useful representations from the training data alone already, and only the classifier needs to be adjusted for the target distribution.

This work questions the conventional usage of the last layer for OOD generalization, providing an in-depth analysis across various distribution shifts and model architectures. We examine both few-shot OOD generalization, where some OOD samples are available for training linear classifier, and zero-shot OOD generalization, which requires no target examples. Our studies reveal that earlier-layer representations often yield superior OOD generalization in both scenarios. Figure 1 illustrates this across multiple datasets. For instance, on CMNIST, using earlier layers improves zero-shot OOD accuracy by 7% and few-shot OOD accuracy by 12% compared to retraining the last layer. The effect is even more pronounced for CelebA, where we observe substantial gains of 20% for zero-shot OOD generalization using earlier layers. These results consistently demonstrate the advantages of using earlier-layer representations for OOD generalization tasks.

We advocate the use of earlier-layer representations for a few critical benefits in practice. First, we show that they often exhibit better generalization performance when tuned on the target distribution; this also extends to cases where the number of samples from the target distribution is small (§4.2). Second, the benefits remain even when no OOD data is used for training the probes, only for using them in the model-selection step (§4.3).

Our contributions are as follows: (1) We establish that earlier-layer representations often outperform last-layer features in terms of (1) few-shot and (2) zeros-shot transfer capabilities, even when no OOD supervision is available. (3) We provide evidence that intermediate layer features are generally less sensitive to distribution shifts than those from the final layer, offering new insights into feature utility for enhancing model robustness.

## 2 RELATED WORK

**OOD generalization and spurious correlations**: It has been reported that spurious correlations in training data degrade the generalizability of learned representations, particularly on the out-of-distribution (OOD) data (Arjovsky et al., 2020; Ruan et al., 2022; Hermann & Lampinen, 2020), especially on groups within the data that are underrepresented (Sagawa et al., 2020a; Yang et al., 2023) and the number of groups is large (Li et al., 2023; Kim et al., 2023). The theory of gradient starvation further suggests that standard SGD training may preferentially leverage spurious correlations (Pezeshki et al., 2021), especially when the spurious cue is simple (Scimeca et al., 2022). Other works have focused on the conceptual ill-posedness of OOD generalization without any information about the target distribution (Ruan et al., 2022; Bahng et al., 2020; Scimeca et al., 2022).

Simplicity bias, where networks favor easy-to-learn features, is influenced by the breadth of solutions exploiting such features compared to those utilizing more complex signals (Geirhos et al., 2020; Valle-Pérez et al., 2018; Scimeca et al., 2022). It has been observed that DNN models tend to perform well on average but worse for infrequent groups within the data (Sagawa et al., 2020a); this is especially exacerbated for overparameterized models (Sagawa et al., 2020b; Menon et al., 2020).

**Last layer-retraining**: Recent studies have argued that last-layer representations already contain valuable information for generalization beyond the training distribution (Kirichenko et al., 2023; Izmailov et al., 2022; LaBonte et al., 2023). These approaches suggest retraining the last layer weights (using the features from the penultimate layer) with either target domain data or using group annotations from large language models (Park et al., 2023). In this work, we challenge the underlying assumption that the *penultimate layer* encapsulates all pertinent information for OOD generalization.

Our analysis indicates that earlier-layer representations are often far more useful for OOD generalization; we even show that earlier-layer representations *without fine-tuning* on the target distribution often fare competitively with the last-layer retraining *on the target distribution*.

**Exploiting and analyzing intermediate layers.** Intermediate layers have been employed for various purposes, from predicting generalization gaps (Jiang et al., 2019), to elucidating training dynamics

(Alain & Bengio, 2018), to enhancing transfer and few-shot learning (Evci et al., 2022; Adler et al., 2021), to enhancing transferablity of adversarial examples (Huang et al., 2020), and adjusting models based on distribution shifts (Lee et al., 2023). The effectiveness of intermediate layers can be connected to the properties of neural network dynamics Yosinski et al. (2014). For example, the intrinsic dimensionality of intermediate layers has been shown to increase in the earlier layers and then decrease in the later layers (Ansuini et al., 2019; Recanatesi et al., 2019), suggesting a rich and diverse set of features in the intermediate layers that can be leveraged for generalization. Neural collapse (Papyan et al., 2020), a phenomenon where the representations of class features collapse to their mean classes in the last layer, points to a reason for the difficulty of transferring features from the last layer. Neural collapse in intermediate layers is observed, but it is less severe (Li et al., 2022; Rangamani et al., 2023). Recent work has also explored the utility of intermediate layers in distinct contexts, such as generalization to new class splits Gerritz et al. (2024); Dyballa et al. (2024), and has arrived at conclusions similar to ours: that the last layer does not always generalize best. Additionally, Masarczyk et al. (2023) observe a complementary phenomenon termed the tunnel effect, in which later layers compress linearly separable representations created by initial layers, thus degrading OOD performance.

Our work differs by examining distribution shifts within individual datasets, rather than generalization across datasets or tasks. Unlike prior studies, we investigate how intermediate layers generalize under controlled in-dataset shifts, even when training and testing involve similar visual variations.

## 3 APPROACH

This section introduces the preliminary concepts and notation as well as our approach to OOD generalization using intermediate-layer representations.

### 3.1 TASK

We consider the classification task of mapping inputs $\mathcal{X}$ to labels $\mathcal{Y}$. We present an overview of the training and evaluation stages of the model in Table 1. A deep neural network (DNN) model is first trained on a training dataset $\mathcal{D}_{\text{train}}$. The DNN is further adapted to the task through the Intermediate Layer Classifiers (ILCs) training (§3.2) on the probe-training dataset $\mathcal{D}_{\text{probe}}$. Afterwards, any model selection or hyperparameter search is performed over the validation set $\mathcal{D}_{\text{valid}}$. Finally, the model is evaluated on the test set $\mathcal{D}_{\text{test}}$.

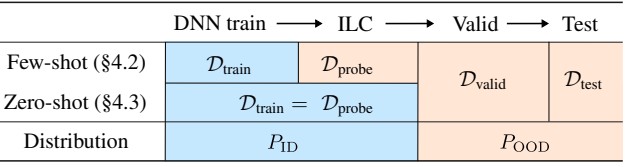

Table 1: **Experimental setups.** We consider settings where an entire deep neural network (DNN) is first trained on $\mathcal{D}_{\text{train}}$ and one of its layers is adapted further on $\mathcal{D}_{\text{probe}}$ (Intermediate Layer Classifiers; ILC in §3.2). The model is then validated on $\mathcal{D}_{\text{valid}}$ and evaluated on $\mathcal{D}_{\text{test}}$. Depending on the scenario type (few-shot vs zero-shot), either up to $\mathcal{D}_{\text{train}}$ or up to $\mathcal{D}_{\text{probe}}$ is considered to be in-distribution $P_{\text{ID}}$ and the rest to be out-of-distribution $P_{\text{OOD}}$.

In order to simulate the OOD generalization scenario, we introduce two different distributions: $P_{\text{ID}}$ and $P_{\text{OOD}}$, respectively denoting in-distribution (ID) and OOD cases. The training of any DNN is performed over ID: $\mathcal{D}_{\text{train}} \sim P_{\text{ID}}$. The validation is performed over OOD: $\mathcal{D}_{\text{valid}} \sim P_{\text{OOD}}$. The usage of a few OOD samples for validation is a standard practice in OOD generalization literature (Gulrajani & Lopez-Paz, 2020; Sagawa et al., 2020a; Izmailov et al., 2022) and we adopt this framework for a fair comparison. Likewise, the final evaluation is performed on OOD: $\mathcal{D}_{\text{test}} \sim P_{\text{OOD}}$. We consider multiple variants of OOD datasets; we discuss them in greater detail in §4.1.

For the ILC training step, we consider two possibilities: (1) **few-shot** where $\mathcal{D}_{\text{probe}} \sim P_{\text{OOD}}$ are $K$ OOD samples per class and (2) **zero-shot** where $\mathcal{D}_{\text{probe}} \sim P_{\text{ID}}$ are trained over the ID set. The last-layer retraining framework (Kirichenko et al., 2023) corresponds to the few-shot learning scenario, where the last layer is retrained with a $K$ OOD samples per class. Our research question for the few-shot scenario is whether intermediate representation yields a better OOD generalization compared to the last-layer retraining paradigm. For the zero-shot scenario, we venture into a more audacious research question: is it possible to train an intermediate representation with ID samples to let it generalize to OOD cases?

## 3.2 INTERMEDIATE LAYER CLASSIFIERS (ILCs)

In this section, we propose the framework for training Intermediate Layer Classifiers (ILC). We start with the necessary notations and background materials.

Let function $f$ be a deep neural network (DNN) classifier with $L$ layers. We denote the $l$-th layer operation as $f_l$, such that $f \equiv f_L \circ f_{L-1} \circ \ldots \circ f_1$, a composition of $L$ functions. The last layer of the network $f_L$ is a linear classifier. We refer to the output of the $l$-th layer for an input $\mathbf{x}$ as the $l$-**th layer representation**, denoted as $\mathbf{r}_l(\mathbf{x}) := (f_l \circ \cdots \circ f_1)(\mathbf{x}) \in \mathbb{R}^{d_l}$, where $d_l$ denotes the output dimension at layer $l$. For $l < L$, we refer to $f_l$ as an **intermediate layer** and $\mathbf{r}_l(\mathbf{x})$ as an **intermediate representation**.

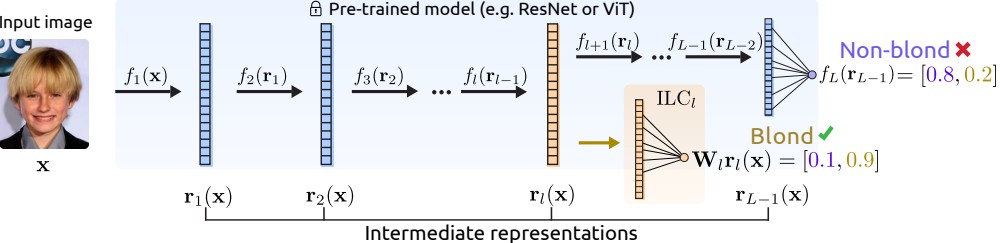

Figure 2: **Intermediate Layer Classifiers (ILC).** Given a frozen pre-trained model like a ResNet or a ViT, we train a linear probe on an intermediate layer representation at intermediate layers (here, we show this process only at layer $l$). The composition of $l^{\text{th}}$ layer feature extractor and the intermediate layer classifier (ILC) is the final classifier. We shorthand $\mathbf{r}_l(\mathbf{x})$ as $\mathbf{r}_l$ for brevity.

We primarily use ResNets (He et al., 2015) for convolutional neural networks and vision transformers (ViTs) (Dosovitskiy et al., 2021) for non-CNN architectures, while other architectures are also used when open-source implementations are available. A ResNet layer, for example, consists of convolutions, ReLU activation, batch normalization, and residual connections, leading to $L = 8$ layers in total, excluding the classification head. A ViT layer is an encoder block composed of multi-head attention (MHA) and a multi-layer perceptron (MLP). The number of layers for ViT models varies depending on the dataset used. All the architectures used are detailed in Appendix A.1.

**Intermediate Layer Classifiers (ILCs).** We introduce Intermediate Layer Classifiers (ILCs) to address the specific challenge of OOD tasks. While traditional linear probes (Alain & Bengio, 2018) are typically used to analyze learned representations across a model, ILCs serve a different purpose: they are applied to intermediate layers and trained specifically to perform OOD classification. An ILC at layer $l$ is an affine transformation that maps the representation space $\mathbb{R}^{d_l}$ to logits:

$$\text{ILC}_l(\mathbf{x}) := \mathbf{W}_l \mathbf{r}_l(\mathbf{x}) + \mathbf{b}_l, \quad (1)$$

where $\mathbf{W}_l \in \mathbb{R}^{|\mathcal{Y}| \times d_l}$ and $\mathbf{b}_l \in \mathbb{R}^{|\mathcal{Y}|}$. The ILCs for $1 \leq l \leq L-1$ are trained on the dataset $\mathcal{D}_{\text{probe}}$ (§3.1), using data drawn from ID for the *zero-shot* scenario ($\mathcal{D}_{\text{probe}} \sim P_{\text{ID}}$) and OOD for the *few-shot* scenario ($\mathcal{D}_{\text{probe}} \sim P_{\text{OOD}}$).

**Algorithm 1** Training Intermediate Layer Classifiers (ILCs)

1: **for** $1 \leq l \leq L-2$ **do**
2:     Initialize weights $\mathbf{W}_l$ and biases $\mathbf{b}_l$ for ILC$_l$
3: **end for**
4: **for** $(\mathbf{x}, y) \sim \mathcal{D}_{\text{probe}}$ **do**
5:     **for** $1 \leq l \leq L-2$ **do**
6:         $\mathbf{r}_l(\mathbf{x}) = (f_l \circ \cdots \circ f_1)(\mathbf{x})$
7:         $\hat{\mathbf{y}}_l = \text{ILC}_l(\mathbf{x}) = \mathbf{W}_l \mathbf{r}_l(\mathbf{x}) + \mathbf{b}_l$
8:     **end for**
9:     $\ell = \sum_{l=1}^{L-1} \text{CE}(\hat{\mathbf{y}}_l, y)$
10:     Update parameters $\phi$ using $\ell$
11: **end for**

We illustrate the data flow of using ILCs conceptually in Figure 2. Last-layer retraining methods (Kirichenko et al., 2023; Izmailov et al., 2022; LaBonte et al., 2023; Kang et al., 2020) can be considered a special case of the ILC framework, where only the final layer $L$ is adapted on $\mathcal{D}_{\text{probe}}$ for OOD tasks, corresponding to using ILC$_{L-1}$ in our framework. Algorithm 1 illustrates the training process of ILCs.

**Layer Selection.** We choose the layer $l^\star$ with the best ICL accuracy on the validation set $\mathcal{D}_{\text{valid}}$ from $l \leq L-2$. We restrict the layer selection to layers up to $L-2$ to distinguish the results from the last-layer retraining approach. The best layer is chosen based on the best-performing hyperparameters in the search space $\mathcal{H}$ (detailed in Appendix A.2.2) for each layer. Using a few OOD samples for making design choices is one of the common practices in benchmarking OOD generalization (Kirichenko et al., 2023; Sagawa et al., 2020a).

**Inference.** For the selected layer $l^\star \leq L - 2$, the whole model is reduced to a smaller network with $l^\star$ layers. Pseudocode for the inference process is provided in Algorithm 4 in the Appendix.

## 4 EXPERIMENTS

In this section, we verify the effectiveness of ILCs in §3.2 for out-of-distribution (OOD) generalization. In particular, we compare their performance against the popular last-layer retraining approach. We introduce the dataset and experimental setups in §4.1. We report results under two scenarios: few-shot (§4.2) and zero-shot (§4.3) cases, respectively referring to the availability and unavailability of OOD data for the ILC training.

### 4.1 DATASETS AND EXPERIMENTAL SETUP

We introduce datasets and a precise experimental setup for simulating and studying OOD generalization. As introduced in the task section (§3.1), we need two distributions $P_{\text{ID}}(X, Y) \neq P_{\text{OOD}}(X, Y)$ for the study. We perform an extensive study over 9 datasets, covering various scenarios, including subpopulation shifts, conditional shifts, noise-level perturbations, and natural image shifts. Detailed definitions of shift types and corresponding datasets are listed in Table 2 below.

Table 2: **Distribution shift types and datasets.** Datasets were selected based on the availability of distribution shifts and their compatibility with publicly available pre-trained model weights.

| Shift type | Description | Datasets |
|---|---|---|
| Conditional | The conditional distribution $P(Y\|X)$ shifts: $P_{\text{train}}(Y\|X) \neq P_{\text{test}}(Y\|X)$. | CMNIST (Arjovsky et al., 2020; Bahng et al., 2020) |
| Subpopulation | Given multiple groups within a population, the ratios of groups change across distribution. | CelebA (Liu et al., 2015), Waterbirds (Sagawa et al., 2020a), Multi-CelebA (Kim et al., 2023) |
| Input noise | Different types of input noise are applied for test samples. | CIFAR-10C, CIFAR-100C (Hendrycks & Dietterich, 2019) |
| Natural image | Test images have different styles from the training images. | ImageNet-A, ImageNet-R, ImageNet-Cue-Conflict, ImageNet-Silhouette (Hendrycks et al., 2021b;a; Geirhos et al., 2022) |

In §3.1, we have introduced the few-shot and zero-shot settings for the OOD generalization. Below, we explain how we adopt each dataset for the required data splits, $\mathcal{D}_{\text{train}}$, $\mathcal{D}_{\text{probe}}$, $\mathcal{D}_{\text{valid}}$, and $\mathcal{D}_{\text{test}}$. Detailed information on the datasets and their splits is provided in Appendix A.2.

**Training split $\mathcal{D}_{\text{train}}$.** For all datasets, we assume DNN models were trained on the given training split.

**Probe-training split $\mathcal{D}_{\text{probe}}$.** For the zero-shot setting, we use the $\mathcal{D}_{\text{train}}$ split. For the few-shot setting, we use a subset of the OOD splits of each dataset.

**Validation split $\mathcal{D}_{\text{valid}}$.** In all settings, we use the original held-out validation set whenever available in the datasets (Waterbirds, CelebA, MultiCelebA, ImageNet). When unavailable (CMNIST, CIFAR-10C, CIFAR-100C), we use a random half of the test splits of the datasets.

**Test split $\mathcal{D}_{\text{test}}$.** In all settings, we use the original test set. When half of it was used for validation due to a lack of a validation split, then we use the other half for evaluating the models.

**Evaluation metrics.** We use the accuracy on the test set as the main evaluation metric. For datasets with subpopulation shifts, we use the worst-group accuracy (WGA), defined as the minimal accuracy over different sub-populations of the dataset (Sagawa et al., 2020a).

**DNN model usage and selection.** We exclusively use publicly available pre-trained model weights trained on a specific dataset relevant to our experiments. Importantly, we only use frozen representations from these networks and *do not fine-tune any parameters of the DNNs*. We primarily use ViTs and ResNets in our study due to their differing inductive biases. The availability of pre-trained model weights varied, and we aimed to include the most popular and high-performing models within each distribution shift.

## 4.2 RESULTS UNDER THE FEW-SHOT SETTING

We evaluate model performance when a few labeled OOD samples are available for ILC or last-layer retraining. Our goal is to challenge the assumption that the penultimate layer contains sufficient information for OOD generalization (Izmailov et al., 2022; Kirichenko et al., 2023; Rosenfeld et al., 2022) and to inspect the common practice of probing the last layer for this purpose (Zhai et al., 2020). To do so, we compare the effectiveness of intermediate representations with that of the penultimate layer.

### 4.2.1 INFORMATION CONTENT FOR OOD GENERALIZATION AT LAST VERSUS INTERMEDIATE LAYERS

We measure the information content in the last-layer representation versus the intermediate layers by evaluating their accuracy on OOD tasks. To quantify this, we assume a large number of OOD samples, meaning that the entire validation set, as defined in §4.1, is used for $\mathcal{D}_{\text{probe}} \sim P_{\text{OOD}}$.

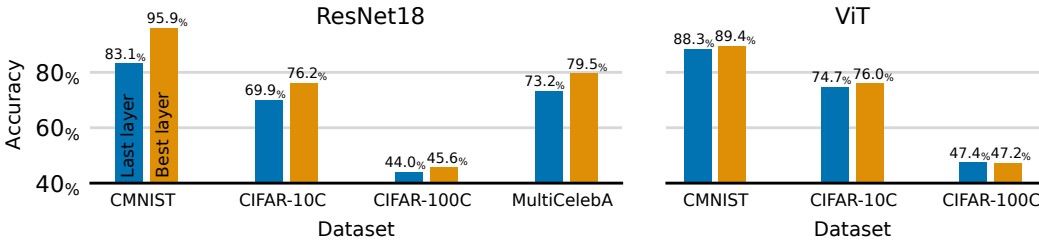

Figure 3: **Information content for OOD generalization in last layer vs intermediate layers.** "Last layer" refers to the OOD accuracy of the last-layer retraining approach (ILC$_{L-1}$); "Best layer" refers to the maximal OOD accuracy among the intermediate layer classifiers (ILC) (ILC$_{l^*}$). For MultiCelebA, we report the worst-group accuracy (WGA).

Fig. 3 shows the OOD accuracies of the last-layer retraining approach and the best performance from the ILC across different layers. Out of the six datasets considered, we observe a general increase in information content when intermediate layers are utilized for OOD generalization instead of the last layers. For ResNets, the performance increments are $(+16.8, +6.3, +1.6, +6.3)$ percentage points on (CMNIST, CIFAR-10, CIFAR-100C, MultiCelebA). ViT has seen smaller or slightly negative increments of $(+1.1, +1.3, -0.2)$ percentage points on (CMNIST, CIFAR-10C, CIFAR-100C). This point is further supported by experiments with non-linear probes (Appendix C.2) and an analysis controlling for feature dimensionality (Appendix C.3), both yielding similar findings.

We conclude that abundant information exists for OOD generalization in the intermediate layers of a DNN. The current practice of utilizing only the last layer representations may neglect the hidden information sources in the earlier layers.

### 4.2.2 OOD DATA EFFICIENCY FOR LAST VERSUS INTERMEDIATE LAYERS

The previous experiment measures the maximal information content at different layers with abundant OOD data to train the probe; here, we consider the data efficiency for OOD generalization at different layers. In practice, it is crucial that a good OOD generalization is achieved with a restricted amount of OOD data. In this experiment, we control the amount of probe training set $\mathcal{D}_{\text{probe}} \sim P_{\text{OOD}}$ of OOD samples with the parameter $\pi \in (0, 1)$ controlling the fraction of OOD data used, compared to the setting in §4.2.1 (corresponding to $\pi = 1.0$). While achieving the best performance is not the primary goal, we also benchmark ILC's results against other methods in Appendix A.5.2.

We illustrate the ILC and last-layer retraining performances on 6 datasets in Fig. 4. For subpopulation shifts (Waterbirds, CelebA, MultiCelebA), we find that training with a smaller amount ($\pi \leq 0.03$) of OOD data leads to a greater empirical advantage of ILCs compared to last-layer retraining with $(+5.7, +3.4, +27.0)$ percentage points. For the other shifts (CMNIST, CIFAR-10C, CIFAR-100C), we observe a consistent benefit of ILC compared to the last-layer retraining. For example, at $\pi = 0.25$, ILC boosts the performance by $(+12, +3.6, +1.0)$ percentage points. ViTs exhibit a similar pattern where ILCs perform better under little OOD data, but the difference is less pronounced (Appendix, Fig. A.5.1).

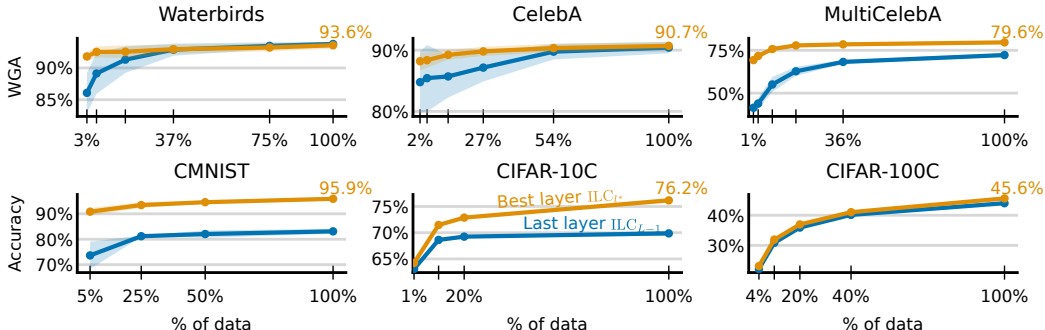

Figure 4: **Accuracies of ILCs and last layer retraining under varying number of OOD samples for ResNets.** Performance of best ILCs and last-layer retraining on subpopulation shifts (first row) and the remaining shifts (second row) using CNN models. We used ResNet50 for Waterbirds and CelebA, and ResNet18 for the remaining datasets.

> **Takeaway from §4.2:** Last layer representations are often sub-optimal for OOD generalization; intermediate layers offer better candidates. The effect is more pronounced when only a small fraction of OOD samples are available to train the linear probe on top.

### 4.3 RESULTS UNDER THE ZERO-SHOT SETTING

We now explore a scenario where *no OOD samples* are available for training the linear probes, but only ID samples are $\mathcal{D}_{\text{probe}} \sim P_{\text{ID}}$. This scenario is intriguing both conceptually and practically. Conceptually, it challenges us to extract features that are effective for OOD generalization using only ID data. This requires leveraging the structure of ID data to identify characteristics that may generalize well to unseen OOD cases. Practically, the assumption that OOD data is unavailable greatly broadens potential application scenarios. We follow the setup outlined in §3.1.

We stress that this zero-shot setting was not considered in previous last-layer retraining methods (Izmailov et al., 2022; LaBonte et al., 2023), which involved re-training the last layer on OOD data.

### 4.3.1 SUBPOPULATION SHIFTS

We illustrate results on subpopulation shifts in Fig. 5. We consider three variations of models: (1) the pre-trained frozen DNN (*Base*), (2) last-layer re-training with the ID data (*Last layer*), and (3) the best ILC after training on ID (*Best layer*). They can be compared as they solve the same task. We also compare these zero-shot results to performant baselines in Appendix A.6.

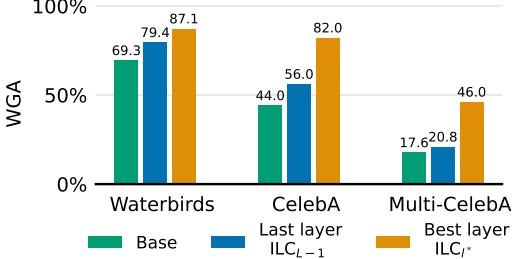

In all three datasets, we observe the relationship: *Base<Last layer<Best layer*. First of all, the relationship *Base<Last layer* is intriguing because just by re-training the last layer of a pre-trained model on the same training set improves

Figure 5: **WGA on OOD data for ID-trained CNNs.** For the explanations for *Base*, *Last layer*, and *Best layer*, refer to the text. ResNet50: Waterbirds, CelebA. ResNet18: MultiCelebA.

the OOD generalization performance. This is a remark out of the paper's scope, but exploring this phenomenon may result in novel insights. Wide gaps are seen for *Last layer<Best layer*, particularly for Waterbirds (79.4%→87.1%), CelebA (56.0%→82.0%) and MultiCelebA (20.8%→46.0%).

We conclude that under subpopulation shifts, intermediate layers can be effectively leveraged using only ID data. In contrast, last-layer retraining may fall short, likely because the features in the final layer are already over-specialized to the training distribution, possibly due to the memorization of minority datapoints (Sagawa et al., 2020a). Our results suggest that intermediate layers may not suffer from this issue to the same extent.

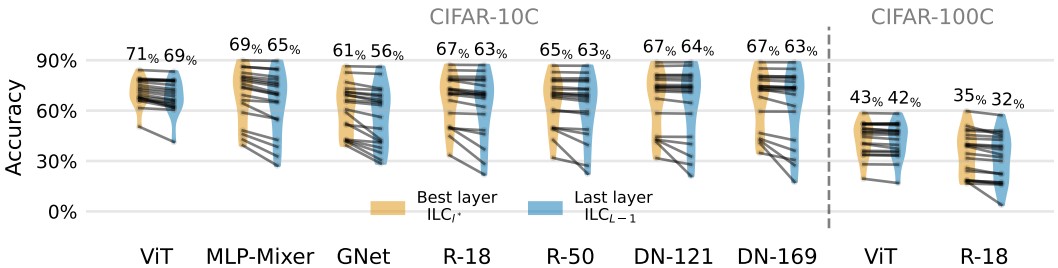

Figure 6: **CIFAR-10C and CIFAR-100C results for zero-shot ILC and last-layer retraining.** We compare the best layer and last layer performances on multiple model architectures. Since there are 19 noise types, we plot the distribution of accuracies with violin plots showing the kernel density estimation outputs. We link results for individual noise types across best layer and last layer results with solid lines.

### 4.3.2 INPUT PERTURBATION SHIFTS

We show the zero-shot performance of ILC and last-layer retraining on OOD shifts with input perturbations. Results for CIFAR-10C and CIFAR-100C are provided in Fig. 6. We report the performance of multiple models with publicly available weights, including ResNet-18 and ViT. Detailed descriptions of these models can be found in Appendix A.1.

On CIFAR-10C, all models show +2%p to +5%p improvements in mean accuracies when the best layer is used, instead of the last layer. ViT performs best under the ILC framework, reaching an average accuracy of 71%. For CIFAR-100C, the accuracy gains are +3%p for ResNet-18 and +1%p for the ViT model. These results consistently demonstrate the benefits of utilizing intermediate-layer representations for improving OOD generalization across different model architectures.

### 4.3.3 STYLE SHIFTS

We present the zero-shot relative average accuracies between ILC and last-layer retraining in Fig. 7. For cue-conflict and silhouette style shifts, the average accuracy boosts for ILC are +2 percentage points. For Imagenet-A and ImageNet-R, the average relative accuracies are +0.91 and +0.34 percentage points, respectively. The classes in silhouette and cue-conflict datasets are more abstract and require the model to rely on other cues, such as shape, which could explain the higher relative improvements in those cases.

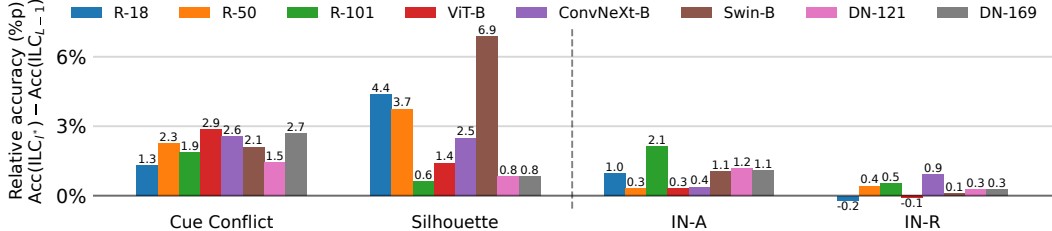

Figure 7: **Performance boost for intermediate layers on ImageNet variants for zero-shot setting.** We show the performance of the best intermediate layer relative to the last layer.

> **Takeaway from §4.3:** OOD performance can be improved using ILCs only using ID data as long as a held-out dataset is available for model selection.

## 5 ANALYSIS

In this section, we analyze the contributing factors behind the improved generalization performance of the ILCs shown in the previous section. We analyze the impact of depth and corresponding distances between embeddings of in-distribution and out-of-distribution samples.

## 5.1 Impact of Depth

We inspect the contribution of depth to the generalizability of linear probes trained on intermediate representations. We consider both few-shot and zero-shot scenarios, according to the availability of OOD data (§4). For the few-shot case, we consider the oracle model selection case due to training ILCs on the whole $\mathcal{D}_{\text{valid}}$. We report the performances of ILCs trained on all layers in Fig. 8.

We observe two key patterns:

(1) **The pen-penultimate layer consistently outperforms the penultimate layer.** Across all datasets, the pen-penultimate layer (layer 7) achieves higher OOD accuracy than the penultimate layer (layer 8) in both few-shot and zero-shot scenarios. For instance, in the Waterbirds dataset, the 7th layer (pen-penultimate) attains a worst-group accuracy (WGA) of 94.0% in the few-shot scenario and 86.7% in the zero-shot scenario, compared to the 8th layer's (penultimate) 93.9% and 79.4%, respectively. This indicates that representations from the pen-penultimate layer generalize better under distribution shifts than those from the penultimate layer.

(2) **The best-performing layer varies across datasets.** The optimal layer for ILCs depends on the distribution shift and dataset. In CelebA and Multi-CelebA, earlier layers outperform the penultimate layer. For example, the 5th layer yields 80.6% WGA in the

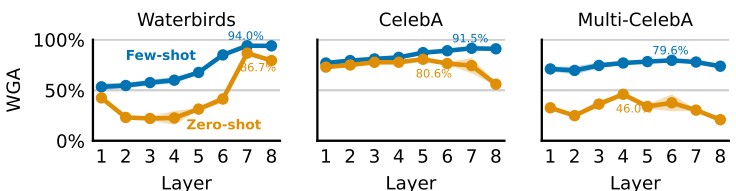

Figure 8: **OOD accuracies across layers.** We consider subpopulation shifts (Waterbirds, CelebA, and Multi-CelebA) in cases when OOD data is available ('Few-shot') and when it is not ('Zero-shot').

zero-shot scenario for CelebA, and the 6th layer achieves 79.5% WGA in the few-shot scenario for Multi-CelebA. We observe similar patterns of depth impact in other datasets like CIFAR-10 and CIFAR-100; detailed results for these datasets using ResNet-18 models are provided in the Appendix (see Fig. 17).

> **Takeaway from §5.1:** When training ILCs on a single layer, classifiers built on intermediate representations—particularly the pen-penultimate layer—often outperform those built on the penultimate layer.

## 5.2 Feature sensitivity in intermediate and penultimate layers

We observed that in §4.2.1 the few-shot setting, ILCs consistently outperform last-layer retraining, which relies on features from the penultimate layer. This observation suggests that the information content in penultimate layers is smaller compared to the feature in the intermediate layers.

We hypothesize that the increased performance is not only because these features are inherently more informative about the factors of variation within the data but also because they exhibit greater invariance to distribution shifts compared to those in the penultimate layer. According to this hypothesis, for example, if an image is perturbed by a small amount of noise, the features in the penultimate layer may change significantly, while the features in the intermediate layers may remain more stable. This would explain ILCs' better performance in the zero-shot setting (§4.3): classifiers built on top of ID features would transfer better if the features derived from OOD data were similar to those derived from ID data.

In this section we test this hypothesis. We focus our analysis on subpopulation shifts here; similar results hold for input-level shifts (Appendix B.2.1).

**Intermediate layers are less sensitive to distribution shifts.** We test the hypothesis that intermediate layers are less sensitive to distribution shifts. We define an intuitive global metric to compare how far test data points deviate from the training data at each layer. This is done by comparing the average distances between points in the training set and those in the test set. The ratio of these distances gives a sense of how much a layer "notices" the shift. This helps us understand which layers remain stable under distribution shifts and which are more sensitive.

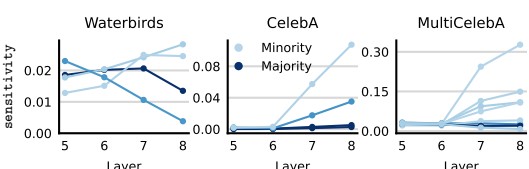 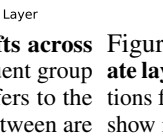 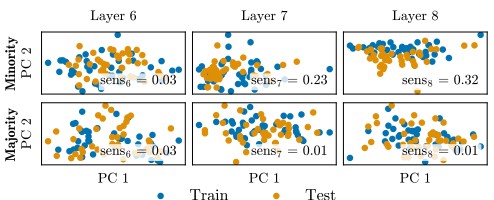

Figure 9: **Sensitivity to subpopulation shifts across layers.** The "Minority" refers to the least frequent group in the dataset (dark blue). The "Majority" refers to the most frequent group (light blue). Groups in between are colored in shades of blue.

Figure 10: **ID and OOD separation in intermediate layers (ResNet18 on MultiCelebA).** PCA projections for minority (top) and majority (bottom) groups show increasing test-train separation in earlier layers, with layer 8 showing the largest separation for one minority group.

Concretely, we measure the mean pairwise distance between the training points and testing points belonging to the same group $g$ (e.g. a group of blond males in CelebA), and normalize this quantity by the mean pairwise distance between the probe points within that group. The datasets $\mathcal{D}_{\mathrm{probe}_g}$ and $\mathcal{D}_{\mathrm{test}_g}$ refer to the training and testing sets from group $g$, respectively. We define the mean pairwise distance $\mathtt{dist}_l(\mathcal{D}_1, \mathcal{D}_2)$ and the sensitivity score $\mathtt{sens}_l$ for layer $l$ as:

$$\mathtt{dist}_l(\mathcal{D}_1, \mathcal{D}_2) := \frac{1}{|\mathcal{D}_1||\mathcal{D}_2|} \sum_{i,j} \left\| \mathbf{r}_l(\mathcal{D}_1^{(i)}) - \mathbf{r}_l(\mathcal{D}_2^{(j)}) \right\|_2^2, \quad \mathtt{sens}_l := \left| 1 - \frac{\mathtt{dist}_l(\mathcal{D}_{\mathrm{probe}_g}, \mathcal{D}_{\mathrm{test}_g})}{\mathtt{dist}_l(\mathcal{D}_{\mathrm{probe}_g}, \mathcal{D}_{\mathrm{probe}_g})} \right|,$$

where $\mathbf{r}_l(\mathcal{D}_1^{(i)})$ and $\mathbf{r}_l(\mathcal{D}_2^{(j)})$ denote the representations of the individual samples $i$ and $j$ from the datasets at layer $l$; The sensitivity score $\mathtt{sens}_l$ of 0 indicates that the test points are as far from the training points as the training points are from each other. A high sensitivity score indicates that, on average, the test representations are more separated from the training points, reflecting greater sensitivity to distribution shifts.

We illustrate the sensitivity scores between ID and OOD data for the subpopulation shifts in Fig. 9. We observe two key patterns:

(1) **Minority groups are more sensitive to distribution shifts.** Features extracted from OOD data (i.e., minority groups) tend to have higher sensitivity scores compared to those from majority groups, particularly at deeper layers. This indicates that underrepresented samples from minority groups are mapped further away from the ID training samples in feature space compared to majority groups.

(2) **Intermediate layers are less sensitive to shifts compared to the penultimate layer for minority groups.** Across both CelebA and MultiCelebA, we observe that intermediate layers exhibit significantly lower sensitivity scores, often collapsing to a single low value, while the penultimate layer remains more sensitive to subpopulation shifts. This suggests that intermediate layers are more invariant to distribution shifts, particularly for underrepresented groups, while the penultimate layer tends to overemphasize these shifts.

In Fig. 10 we illustrate the separation between ID and OOD datapoints qualitatively. The projections of minority and majority group datapoints show that for the minority group, highest separation between testing and training points occurs at the penultimate layer, with earlier layers clustering points closer together. The majority group shows less separation across layers.

> **Takeaway from §5.2:** The reason ILCs outperform last-layer retraining on OOD generalization tasks may be due to the robustness of intermediate layers to distribution shifts.

# 6 CONCLUSION

While prior work has shown that last-layer retraining can be sufficient for certain OOD generalization tasks, our results indicate that this approach can sometimes fall short. We demonstrate that intermediate-layer representations are less sensitive to distribution shifts than the penultimate layer. By introducing ILCs, which capitalize on this reduced sensitivity to distribution shifts, we provide a stronger baseline for zero-shot and few-shot learning. These findings suggest a need to reassess the focus on penultimate-layer features and leverage the stability of intermediate layers for more effective generalization in complex, real-world scenarios.

## 7 ACKNOWLEDGEMENTS

We thank the anonymous reviewers for their valuable feedback, and the International Max Planck Research School for Intelligent Systems (IMPRS-IS) for supporting Arnas Uselis. This work was supported by the Tübingen AI Center. We also acknowledge Alexander Rubinstein and Ankit Sonthalia for their helpful insights.

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

# A  DETAILS ON MAIN RESULTS

## A.1  COMPOSITION OF USED MODELS

We detail the layers of the main models considered throughout this study. We use ResNet variants from the TorchVision library (maintainers & contributors, 2016). The models from the ResNet architecture family can be represented as a composition of blocks, e.g., for the ResNet-50 model, we have:

$$\texttt{ResNet-50} := \texttt{Block}^0 \circ \texttt{Block}^2 \circ \cdots \circ \texttt{Block}^8.$$

More specifically, we can describe the structure of ResNet-18 and ResNet-50 as follows:

$$\begin{aligned}
\texttt{ResNet-18} := &\texttt{Conv}(64, 7, 2) \to (\texttt{BatchNorm} \to \texttt{ReLU}) \to \texttt{MaxPool}(3, 2) \to \\
&\texttt{ResBlock}(64, 2) \to \texttt{ResBlock}(128, 2) \to \\
&\texttt{ResBlock}(256, 2) \to \texttt{ResBlock}(512, 2) \to \texttt{AdaptiveAvgPool},
\end{aligned}$$

$$\begin{aligned}
\texttt{ResNet-50} := &\texttt{Conv}(64, 7, 2) \to (\texttt{BatchNorm} \to \texttt{ReLU}) \to \texttt{MaxPool}(3, 2) \to \\
&\texttt{ResBlock}(64, 3) \to \texttt{ResBlock}(128, 4) \to \\
&\texttt{ResBlock}(256, 6) \to \texttt{ResBlock}(512, 3) \to \texttt{AdaptiveAvgPool},
\end{aligned}$$

where:

- $\texttt{Conv}(\text{filters}, \text{kernel size}, \text{stride})$: convolutional layer,
- $\texttt{BatchNorm}$: batch normalization layer,
- $\texttt{ReLU}$: rectified linear unit,
- $\texttt{MaxPool}(\text{pool size}, \text{stride})$: max pooling layer,
- $\texttt{ResBlock}(\text{filters}, \text{number of blocks})$: residual block, consisting of a bottleneck structure with three convolutional layers (1x1, 3x3, 1x1), each followed by a batch normalization layer, and ReLU activations after the first two convolutions, plus a skip connection,
- $\texttt{AdaptiveAvgPool}$: adaptive average pooling layer.

Note that our description of the models excludes the last layer corresponding to the classification head.

### A.1.1  CIFAR-SPECIFIC RESNET MODELS

For CIFAR-10 and CIFAR-100, the ResNet architecture is modified to account for the smaller input size (32x32 instead of 224x224). Key differences are:

1. Removal of the initial 7x7 convolution and 3x3 max pooling,
2. Reduced stride from 2 to 1.

### A.1.2  VIT FOR CIFAR-10

The architecture of the Vision Transformer (ViT) for CIFAR-10 is as follows:

$$\begin{aligned}
\texttt{ViT CIFAR-10} := &\texttt{EmbeddingLayer}(192,\ 384) \to \texttt{EncoderLayer}^0 \to \\
&\texttt{EncoderLayer}^1 \to \texttt{EncoderLayer}^2 \to \texttt{EncoderLayer}^3 \to \\
&\texttt{EncoderLayer}^4 \to \texttt{EncoderLayer}^5 \to \texttt{EncoderLayer}^6 \to \\
&\texttt{EncoderLayer}^7 \to \texttt{EncoderLayer}^8 \to \texttt{EncoderLayer}^9 \to \\
&\qquad\qquad \texttt{CLSExtraction} \to \texttt{Norm}(384)
\end{aligned} \tag{2}$$

Where:

- $\texttt{EmbeddingLayer(in\_features, out\_features)}$: Linear layer with input size 192 and output size 384.

- `EncoderLayer`: Composed of:
  - `LayerNorm(384)`
  - Multi-head Self-Attention (384 dimensions)
  - `MLP(384, 384)` with `GELU` activation
- `CLSExtraction`: Extracts the [CLS] token for classification.
- `Norm(384)`: Layer normalization with feature size 384.
- `FC(384, 10)`: Fully connected layer mapping 384 features to 10 output classes (CIFAR-10).

### A.1.3 ViT FOR CMNIST

The architecture of the Vision Transformer (ViT) for CMNIST is as follows:

$$\texttt{ViT CMNIST} := \texttt{EmbeddingLayer}(147, 384) \rightarrow \texttt{EncoderLayer}^0 \rightarrow \texttt{EncoderLayer}^1 \rightarrow$$
$$\texttt{EncoderLayer}^2 \rightarrow \texttt{EncoderLayer}^3 \rightarrow \texttt{EncoderLayer}^4 \rightarrow$$
$$\texttt{CLSExtraction} \rightarrow \texttt{Norm}(384)$$

## A.2 EXPERIMENTAL SETTING

### A.2.1 DATASET SPLITS IN ZERO- AND FEW-SHOT SETTINGS

Here we outline the construction of datasets and the implementation of zero- and few-shot settings.

**Few-shot setting.** In the few-shot case, we assume the availability of OOD samples for either training ILCs or performing last-layer retraining, $\mathcal{D}_{\text{probe}}$, as well as a validation set $\mathcal{D}_{\text{valid}}$ of OOD samples for selecting the best layer.

We consider two settings: a setting for deriving upper-bound performance of last-layer retraining and ILCs, and a setting where OOD samples for training ILCs is limited. In the first setting we assume oracle model selection process, that is, $\mathcal{D}_{\text{valid}} \leftarrow \mathcal{D}_{\text{test}}$.

CMNIST dataset has an associated OOD testing set. We uniformly sample half of it and assign to to $\mathcal{D}_{\text{probe}}$ and the remaining half to $\mathcal{D}_{\text{test}}$. CIFAR-10C and CIFAR-100C have 19 noise types associated with them. We carry out our experiments for each distribution shift separately, essentially treating each noise level as its own OOD dataset. For each of the datasets, we split them into two equally sized subsets: $\mathcal{D}_{\text{probe}}$ and $\mathcal{D}_{\text{test}}$. We train ILCs and perform last-layer retraining on $\mathcal{D}_{\text{probe}}$, and test them on $\mathcal{D}_{\text{test}}$. For each of Waterbirds, CelebA, and MultiCelebA, there is an associated held-out set $\mathcal{D}_{\text{held-out}}$[1] and testing set $\mathcal{D}_{\text{test}}$. The held-out set takes form of $\mathcal{D}_{\text{held-out}} = \cup_i^G \mathcal{D}_{\text{held-out}}^i$ with samples that have associated group labels. Following (Kirichenko et al., 2023), we take this held-out set and balance it with respect to the number of samples within each group. We achieve this for each dataset by taking the number of samples in the smallest group and subsampling the other groups to match this number using uniform sampling. This results in $\mathcal{D}_{\text{probe}} \leftarrow \cup_i^G \mathcal{D}_{\text{balanced}}^i$, with $|\mathcal{D}_{\text{balanced}}^i| = |\mathcal{D}_{\text{balanced}}^j|$ for $i, j \leq G$, and $\mathcal{D}_{\text{balanced}}^i \subseteq \mathcal{D}_{\text{held-out}}^i$ that we use for training the ILCs. For testing, we use associated test sets that come with these benchmarks $\mathcal{D}_{\text{test}}$, taking form of $\mathcal{D}_{\text{test}} = \cup_i^G \mathcal{D}_{\text{test}}^i$.

In the second setting we consider a scenario where ILCs are trained on $\mathcal{D}_{\text{probe}}^\pi$, where only a fraction of data $\pi \in (0, 1)$ from $\mathcal{D}_{\text{probe}}$ are assigned to $\mathcal{D}_{\text{probe}}^\pi$; $1 - \pi$ proportion of the data from the original $\mathcal{D}_{\text{probe}}$ is used for validation. That is, we divide $\mathcal{D}_{\text{probe}} \leftarrow \mathcal{D}_{\text{probe}}^\pi \cup \mathcal{D}_{\text{valid}}^\pi$ with $|\mathcal{D}_{\text{probe}'}| = \lfloor \pi |\mathcal{D}_{\text{probe}}| \rfloor$, and use $\mathcal{D}_{\text{probe}}^\pi, \mathcal{D}_{\text{valid}}^\pi$ for training and validating the ILCs, respectively. In words, we uniformly sample a proportion of $\pi$ datapoints from $\mathcal{D}_{\text{probe}}$ and assign them to $\mathcal{D}_{\text{probe}}^\pi$, and assign the remaining points to $\mathcal{D}_{\text{valid}}^\pi$.

**Zero-shot setting.** In the zero-shot case, we assume that no OOD data is available for training ILCs or performing last-layer retraining. Instead, we assume that only the ID data on which the DNN model is trained is available for training the ILCs, i.e. $\mathcal{D}_{\text{probe}} \leftarrow \mathcal{D}_{\text{train}}$. For CMNIST, this amounts to using the biased data the DNN model was trained on. For CIFAR-10C and CIFAR-100C,

---

[1]This $\mathcal{D}_{\text{held-out}}$ is usually referred to as a validation set. To avoid confusion with validation set used for model selection, we refer to this set as a held-out set.

$\mathcal{D}_{\text{probe}}$ corresponds to the CIFAR=10 and CIFAR-100 datasets (Krizhevsky, 2000), respectively. For Waterbirds, CelebA, and MultiCelebA, it amounts to the original group-imbalanced dataset used for training the DNN, where group labels are hidden from ILCs. For ImageNet variants, this amounts to either training the ILCs or performing last-layer retraining on the original ImageNet-1K dataset (Russakovsky et al., 2015).

We only assume an OOD set for performing validation. This validation dataset consist of same samples used for training the probes for the few-shot case: for CMNIST it is half of the OOD testing set, for CIFAR it is half of each of the OOD datasets, and for Waterbirds, CelebA, and MultiCelebA, it is the original held-out dataset. Due to the size of OOD sets of ImageNet variants being relatively small, we perform oracle model selection on it.

### A.2.2 HYPERPARAMETER SEARCH

We perform a minimal hyperparameter search over the learning rate and $\ell_1$ regularization strength (similarly to (Kirichenko et al., 2023)) when training ILCs. For the zero-shot case, we tune the learning rate $\eta$ and $\ell_1$ regularization strength according to $(\eta, \ell_1) \in \mathcal{H}_{\text{zero-shot}} = \{10^{-4}, 10^{-3}, 10^{-2}\} \times \{0, 10^{-3}, 10^{-2}\}$. For the few-shot case, we use $(\eta, \ell_1) \in \mathcal{H}_{\text{few-shot}} = \{10^{-4}, 10^{-3}, 10^{-2}\} \times \{0, 10^{-4}, 10^{-3}, 10^{-2}\}$, since we found higher regularization rates to help last-layer retraining.

For all experiments, we use the Adam optimizer (Kingma & Ba, 2017), training ILCs and performing last-layer retraining for 100 epochs. We repeat each experiment at least three times, varying the DNN model's seed, the initialization of the ILCs, and the data splits, depending on the specific setting. Depending on availability, we use either Nvidia 2080 or A100 GPUs.

### A.3 TRAINING AND INFERENCE WITH ILCS

Once the best intermediate layer is determined, inference can be simplified by using only the corresponding ILC, as described in Algorithm 4. Instead of propagating through the entire network, inference at layer $l$ involves computing the intermediate representation at the $l$-th layer and predicting the output based on the ILC of that layer. This enables efficient computation by avoiding unnecessary forward passes through the remaining layers $f_{l+1}, \ldots, f_L$.

---

**Algorithm 2** Inference with ILC for $l$-th layer

1: **Input:** Input sample $\mathbf{x}$, pre-trained network $f$, weights $\mathbf{W}_l$, biases $\mathbf{b}_l$, intermediate layer $l$
2: Compute intermediate representation $\mathbf{r}_l(\mathbf{x}) = (f_l \circ \cdots \circ f_1)(\mathbf{x})$
3: Compute logits $\hat{\mathbf{y}}_l = \mathbf{W}_l \mathbf{r}_l(\mathbf{x}) + \mathbf{b}_l$
4: **Output:** Predicted label $\hat{y} = \arg\max(\hat{\mathbf{y}}_l)$

---

### A.4 ILLUSTRATIVE TOY EXAMPLE ON A CONDITIONAL SHIFT FOR INTEMEDIATE LAYERS UTILIZATION

In the following we illustrate the main ideas of this work in a self-contained manner on Colored-MNIST dataset.

We consider the Colored MNIST (Arjovsky et al., 2020), a dataset with a conditional distribution shift (where $P_{\text{train}}(Y \mid X) \neq P_{\text{test}}(Y \mid X)$, but $P_{\text{train}}(X) = P_{\text{test}}(X)$) constructed using the MNIST (Lecun et al., 1998) dataset that artificially creates spurious correlations during training that are not useful for generalization at test time (detailed in Appendix). The training data consists of colored digit images and binary labels indicating whether the digit is $\geq 5$ or $\leq 4$. Some of the training data is corrupted, where the label is flipped. Digit colors are the spurious cues that are more informative than the digit itself during training. At test time, color is de-correlated with the task label. The aim is to obtain a robust representation that encodes both the digit and color information, such that depending on the distribution shift, either the digit or the color can be adopted.

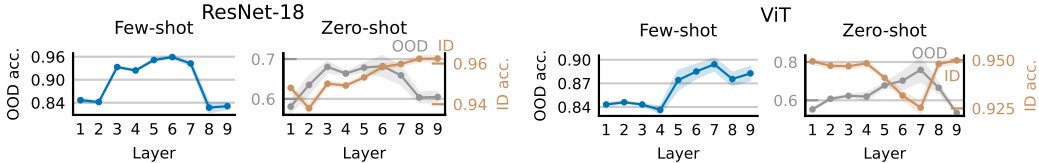

Figure 11: **Qualities of intermediate representations for Colored MNIST.** We show the accuracies of intermediate layers of ResNet-18 and ViT on Colored MNIST. We consider few-shot and zero-shot settings, where the OOD samples are available and unavailable, respectively. Error bars are generated over 3 random seeds. Intermediate representations offer better OOD generalization than the last.

We illustrate the performance of the model across layers when OOD data is not available and when is for ResNet-18 and a ViT (whose compositions are defined above) model with the same depth of 9 layers in Fig. 11; we use 2,000 samples in total for training the probes.

We observe that (1) when OOD data is available ('Few-shot'), the best location to train the linear probe is at an intermediate layer (layer 6 for the ResNet, and layer 7 for ViT) and relying on the last layer gives suboptimal results.

(2) When OOD data is not available ('Zero-shot', noted as 'OOD'), intermediate layers are still better than the last layer; also, the same layers that were optimal when OOD data was available are still optimal when it is not (layers 6 and 7 for ResNet and ViT, respectively).

Finally, (3) the upside of intermediate layers are only for the OOD data — in-distribution data is best classified at the last layer ('Zero-shot', noted as 'ID'), aligned with the findings of previous work (Alain & Bengio, 2018).

## A.5   ADDITIONAL FEW-SHOT RESULTS

### A.5.1   FEW-SHOT RESULTS - VIT

We illustrate accuracies for ViT as a function of fraction of OOD data $\pi$ used for either training the ILCs or performing last layer retraining in Fig. 12.

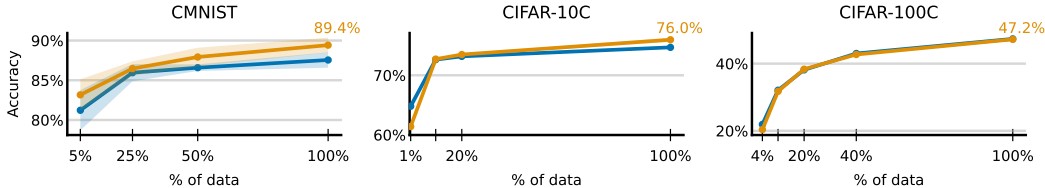

Figure 12: **Accuracies of ILCs and last layer retraining under varying number of OOD samples for ViT.**

### A.5.2   FEW-SHOT: COMPARISON OF ILCS AGAINST OTHER METHODS

We compare ILCs against last-layer retraining (Izmailov et al., 2022) and an approach using multi-object optimization (MOO) by (Kim et al., 2023) in Table 3. We observe that ILCs outperform last-layer retraining and MOO across all datasets while using only a fraction of data.

Table 3: Few-shot results when probes are trained using a varying amount of OOD data and selected using a held-out validation set of OOD data; when 100% of the data is used, the validation is performed on the data the probes are trained on. Our approach matches or exceeds SOTA for subpopulation distribution shifts. MOO results from (Kim et al., 2023).

| Dataset | Data Portion Used (%) | DFR (%) | MOO (%) | ILCs (%) |
|---|---|---|---|---|
| Waterbirds | 75 | $92.2 \pm 0.8$ | N/A | $93.0 \pm 0.8$ |
| Waterbirds | 100 | $92.9 \pm 0.5$ | N/A | $93.0 \pm 0.4$ |
| CelebA | 54 | $87.9 \pm 1.0$ | N/A | $89.2 \pm 0.4$ |
| CelebA | 100 | $88.6 \pm 0.4$ | N/A | $89.7 \pm 0.6$ |
| MultiCelebA | 20 | $73.9 \pm 0.2$ | N/A | $79.0 \pm 0.4$ |
| MultiCelebA | 100 | $76.4 \pm 0.2$ | $77.9 \pm 0.2$ | $79.6 \pm 0.2$ |

### A.6 ZERO-SHOT: COMPARISON OF ILCs AGAINST OTHER METHODS

We compare ILCs against "Just Train-Twice" (JTT) (Liu et al., 2021a), "Correct-N-Contrast" (CnC) (Zhang et al., 2022), and DFR (Kirichenko et al., 2023).

The results are illustrated in Table 4. We note that ILCs outperform DFR and JTT, while only underperforming against CnC, which requires training a specialized model. In contrast, ILCs were trained on the base ERM model without any feature-extractor parameter tuning.

Table 4: Zero-shot results when probes are trained using ID data and selected using a held-out validation set of OOD data. Our approach is competitive with complex methods.

| Dataset | ERM (%) | JTT (%) | CnC (%) | DFR (%) | ILCs (%) |
|---|---|---|---|---|---|
| Waterbirds | 74.9 | 86.7 | 88.5 | 79.4 | 87.1 ± 0.8 |
| CelebA | 46.9 | 81.1 | 88.8 | 56.0 | 82.0 ± 1.1 |

### A.7 CIFAR-10C EXPERIMENTS

CIFAR-C (Hendrycks et al., 2021b) consists of 16 types of noise types: (1) brightness, (2) contrast, (3) defocus, (4) elastic, (5) fog, (6) frost, (7) gaussian, (8) glass, (9) impulse, (10) jpeg, (11) motion, (12) pixelate, (13) saturate, (14) shot, (15) snow, and (16) spatter. This benchmark consists of five levels of noise perturbations of the images in the validation set; we consider only the highest-intensity noise levels for each of the noise types.

**Finetuning on OOD data.** We illustrate each probe's performance across layers when tuning on 1024 points of OOD data for each noise type (Fig. 13) and when tuning on 128 points of OOD data for each noise type (Fig. 14).

**Training on ID data results.** We pre-trained models from the public repository. For the models we train ourselves, in particular: `ViT` (Dosovitskiy et al., 2021) and `MLP-Mixer` (Tolstikhin et al., 2021) we used code following: ViTs for CIFAR public repository and MLP-Mixer for CIFAR public repository.

We consider pre-trained `ViT` (Dosovitskiy et al., 2021), `MLP-Mixer` (Tolstikhin et al., 2021) (`MLP-M`), `GoogleNet` (Szegedy et al., 2014) (`GNet`), `ResNet-{18,50}` (`R-{18, 50}`), and `DensetNet-{121, 169}` (Huang et al., 2018) (`D-{121, 169}`) architectures. Due to the limited availability of other kinds of models pre-trained on CIFAR-100C, we do not include the results for it here.

In Fig. 15 we show the relative performance of the probes across all different noise types on CIFAR-C dataset.

**Using a validation set for selecting the probes.** We illustrate the performance differences when linear probes are selected using a validation set with varying numbers of points in Fig. 16.

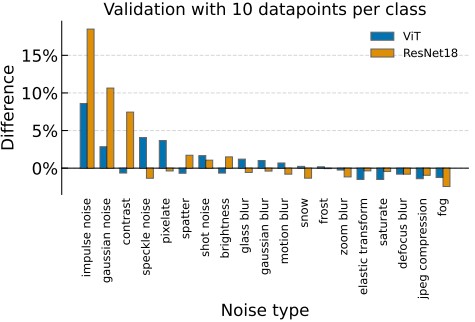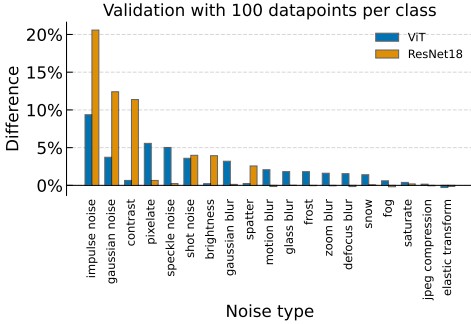

Figure 16: Comparison of performance differences when linear probes are selected on a validation set with varying numbers of points.

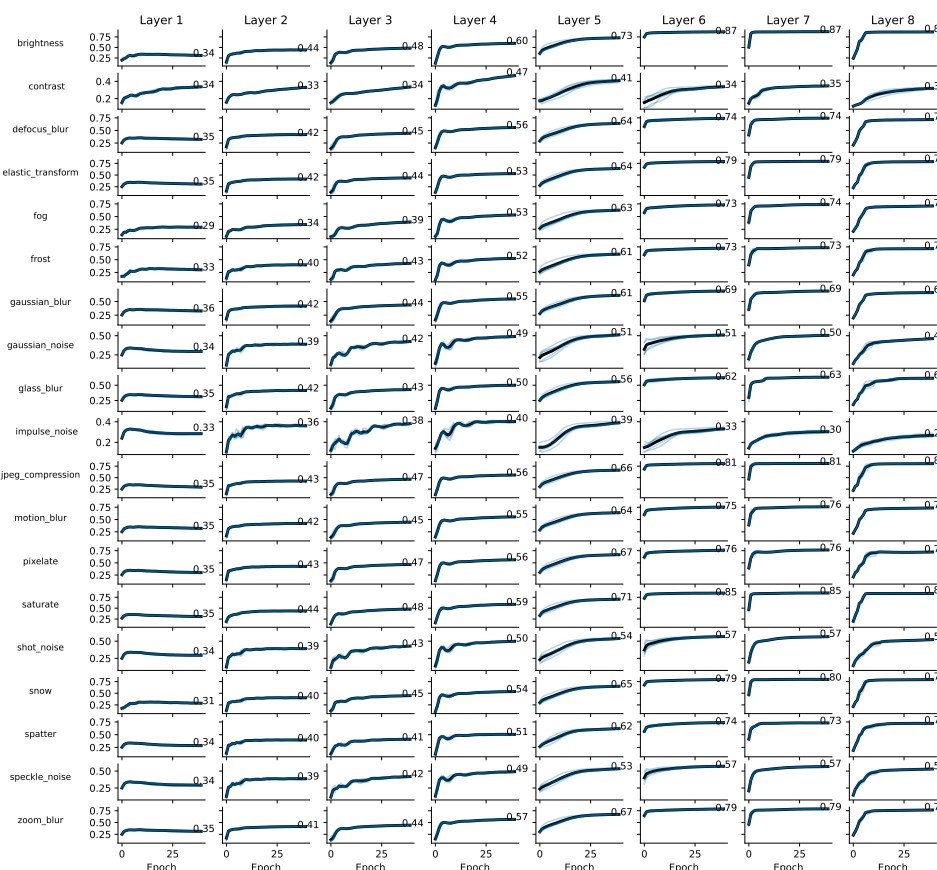

Figure 13: Accuracies of linear probes from last and best-performing layer representations on CIFAR-10C when finetuning on 1024 points of OOD data for each noise type.

## A.8   CIFAR-100

We use pretrained ResNet-18 models on CIFAR-100 from Sonthalia et al. (2024)[2].

## A.9   SUBPOPULATION SHIFTS

For Waterbirds and CelebA experiments, we use a set of 5 pretrained ERM models per dataset trained on different seeds provided in the original (Kirichenko et al., 2023) repository[3]. This model was selected based on one validation worst group accuracy. For MultiCelebA experiments, we use the models from (Kim et al., 2023).

## A.10   "NATURAL" DISTRIBUTION SHIFTS ON IMAGENET-1K

We train probes on ResNets, Swin-B(Liu et al., 2021b)–a base version of the capable ViT variant, and ConvNext-B (Liu et al., 2022), ViT-B-32, and DensNet Huang et al. (2017) architec-

---

[2]Available from `https://github.com/aktsonthalia/starlight`
[3]Available from `https://github.com/PolinaKirichenko/deep_feature_reweighting#checkpoints`.

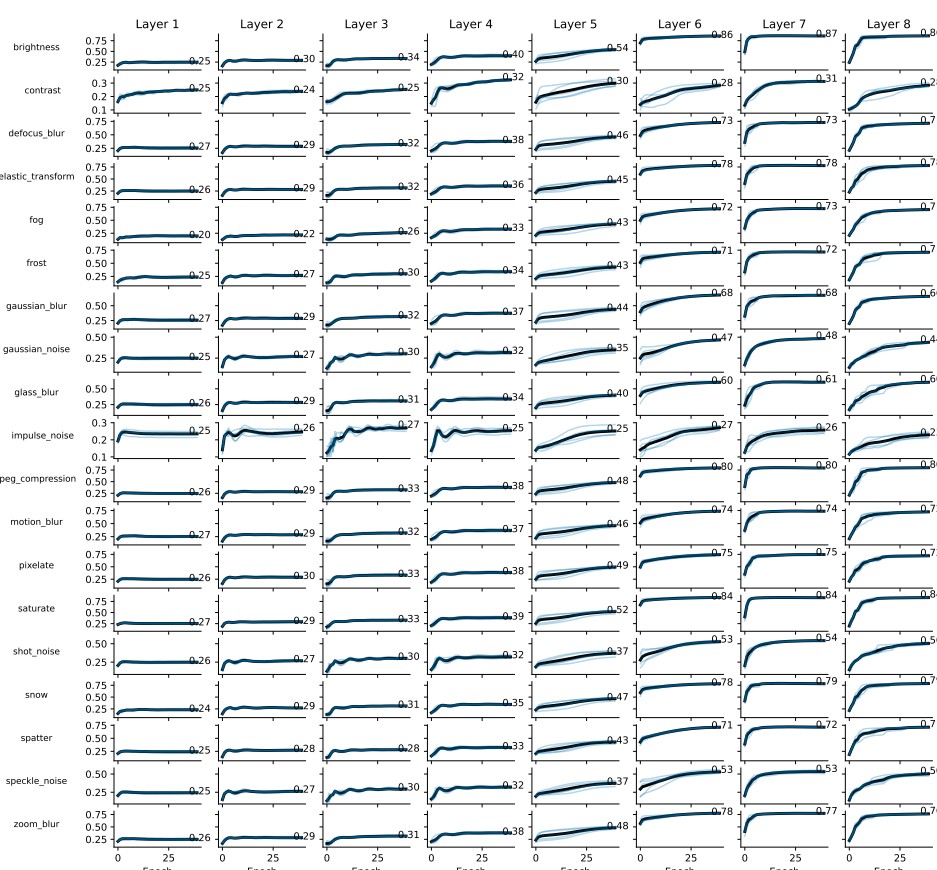

Figure 14: Accuracies of linear probes from last and best-performing layer representations on CIFAR-10C when finetuning on 128 points of OOD data for each noise type.

tures. We divide the considered distribution shifts into two categories: few-class shifts (`Cue Conflict` and `Silhouette`), considering 16 classes each, and many-class shifts (`Imagenet-A` and `Imagenet-R`), considering 200 classes each.

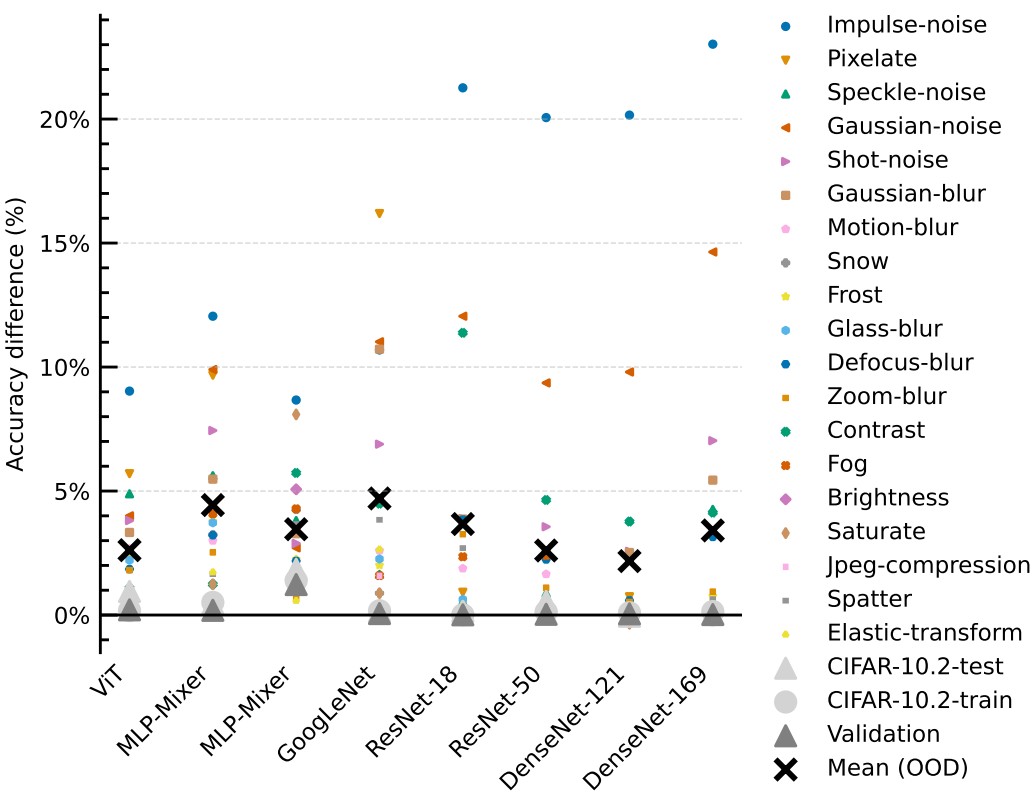

Figure 15: For each model and distribution shift (vertical markers in the plot), relative accuracies for probes across all different noise types on CIFAR-C dataset are illustrated (showing a gap between best performing last layer probe and the itnermediate one). Mean accuracy across datasets is marked as **x** for each model; large gray markers (▲, ●) denote performance on ID datasets.

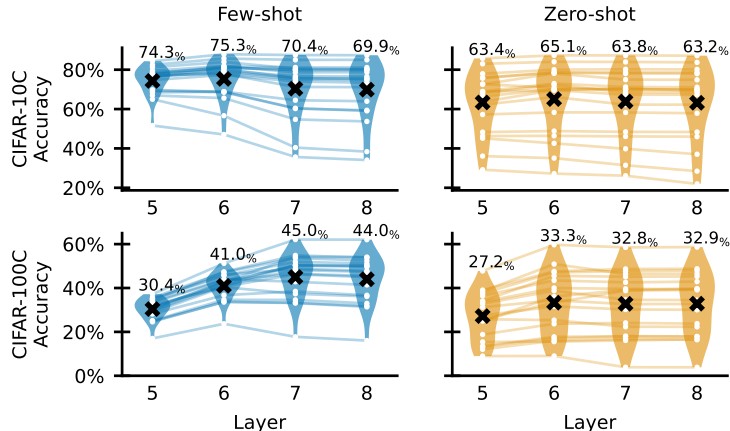

Figure 17: **Impact of Depth on ILC Performance for CIFAR-10 and CIFAR-100.** We evaluate both zero-shot and few-shot settings using ResNet-18. The performance is shown across different layers of the network.

# B  DETAILS ON ANALYSIS

## B.1  IMPACT OF DEPTH

In Fig. B.1 we present additional results on the impact of depth on the performance of ILCs on CIFAR-10 and CIFAR-100 using ResNet-18 model. We observe similar patterns to those in §5.1: (1) the ILC using pen-penultimate layer features ($\text{ILC}_{L-2}(\cdot)$) outperforms last-layer retraining ($\text{ILC}_{L-1}$), and (2) the optimal ILC relies on features from an earlier layer.

### B.1.1  ID AND OOD PERFORMANCE ACROSS LAYERS

In the main text (§5.1), we presented how OOD performances vary across layers in both zero- and few-shot cases. A natural question arises regarding the relationship between OOD performances across layers and ID accuracies. In this section, we conduct experiments to compare ID and OOD accuracies in both settings.

We observe in Fig. 18 that while ID performance generally increases with deeper layers, consistent with previous findings (Alain & Bengio, 2018), OOD performance does not exhibit the same trend. Specifically, the best-performing intermediate layers for zero-shot and few-shot scenarios often lie in the middle or earlier parts of the network, challenging the assumption that the penultimate layer always offers the most robust features for OOD tasks. This reinforces the utility of intermediate layers over the last layer for better OOD generalization.

## B.2  SENSITIVITY ANALYSIS

We explore two ways of measuring sensitivity of features: a global, distance-based way, and a local, neuron-level measure. In the main text (§5.2) we presented results based on a global measure.

### B.2.1  ADDITIONAL RESULTS ON FEATURE SENSITIVITY

The results shown in Figure 19 further reinforce our findings that intermediate-layer representations exhibit lower sensitivity to distribution shifts compared to penultimate-layer representations.

### B.2.2  LOCAL MEASURE OF FEATURE SENSITIVITY

An alternative way of measuring feature sensitivity to OOD shifts is using a local, per-neuron measure. We measure the Total Variation Distance (TVD) between ID and OOD data in intermediate layers. For each dataset and layer, we compare the distributions formed by the representations of ID and OOD data for each feature across the layer. TVD is defined as: $\text{TVD}(P, Q) := \frac{1}{2} \sum_{x \in \text{supp}(P) \cup \text{supp}(Q)} |P(x) - Q(x)|$. We then average the TVDs across all features for each layer. This method provides a measure

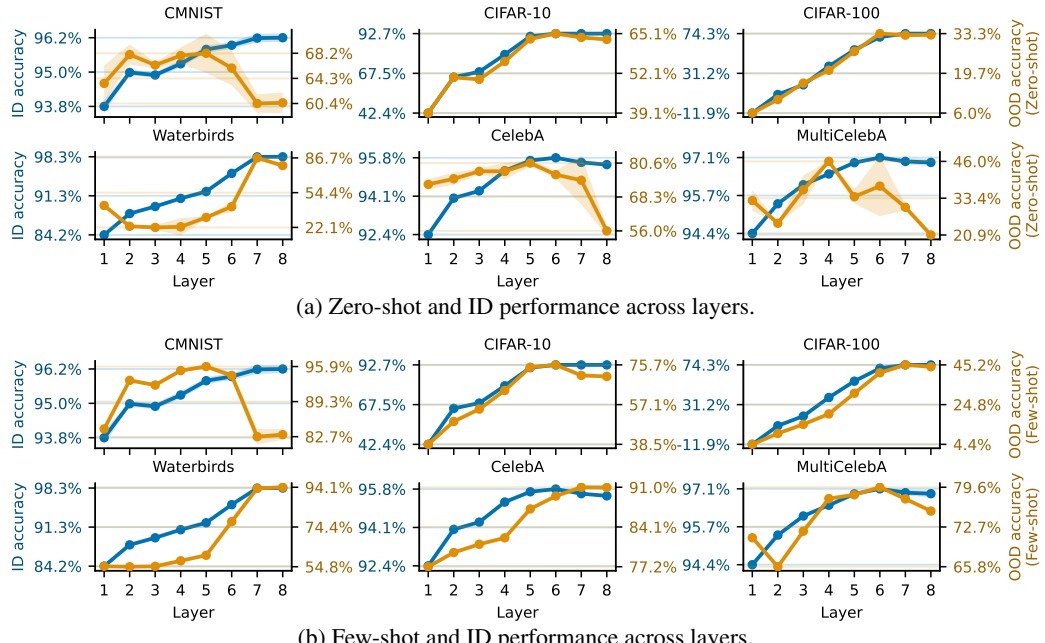

Figure 18: **Performance across layers for ID, and Zero-shot and Few-shot settings.** The comparison of ID and OOD performance across layers. While ID performance generally increases with deeper layers OOD performance does not show the same trend. ResNet models are used.

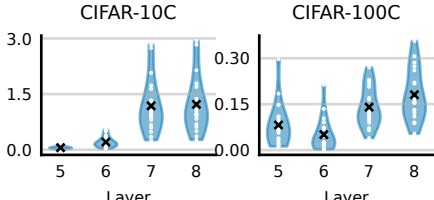

Figure 19: **Sensitivity to input perturbation shifts across layers.** CIFAR-10C and CIFAR-100C feature sensitivity across layers using ResNet-18.

of how much the individual features of the representations vary with the data distribution. A lower TVD indicates that the representations are less affected by distribution shifts, which suggests that classifiers relying on these representations might also be more robust to shifts. This approach allows us to connect the observed transferability of intermediate representations to a quantifiable metric, making the concept of "sensitivity" precise.

Specifically, for each pair $(\mathcal{D}_{\text{ID}}, \mathcal{D}_{\text{OOD}})$ and layer $1 \leq l \leq L$, the layer maps the input space $f_l : \mathbb{R}^{d_{l-1}} \to \mathbb{R}^{d_l}$, producing a flattened representation with $d_l$ features. For each feature $1 \leq i \leq d_l$, we construct distributions $P_{\text{ID}}(f_l(X)_i)$ and $P_{\text{OOD}}(f_l(X)_i)$ by computing the features for ID and OOD data, binning these features into 40 bins, and approximating the distributions using histograms over the union of the ranges of ID and OOD features. Finally, we calculate the TVD between the distributions for each feature and average the TVDs across all features for each layer.

For subpopulation shifts, each dataset exhibits group-level shifts. Group labels allow us to measure shift sensitivity for each group. For example, CelebA consists of four groups (a Cartesian product of two groups: Hair-color and Gender), and for each group $1 \leq g \leq 4$, $\mathcal{D}_{\text{ID}}^g$ is the training data, and $\mathcal{D}_{\text{OOD}}^g$ is the validation data belonging to group $g$. For input-level shifts in CIFAR-10C and CIFAR-100C, for each noise type $1 \leq g \leq 19$, $\mathcal{D}_{\text{ID}}^g$ comprises features from validation set samples (no shift), and $\mathcal{D}_{\text{OOD}}^g$ contains validation data with noise type $g$. To avoid sample imbalance among groups, we consider only the lowest number of samples among the groups. For Waterbirds, CelebA, and MultiCelebA, we consider 67, 1387, and 40 datapoints, respectively. For CIFAR shifts, we consider all 10,000 datapoints.

We illustrate the TVDs between ID and OOD data for the subpopulation shifts and the input-level shifts in Fig. 20. We find that intermediate layers are generally less sensitive to distribution shifts compared to the last layer on subpopulation shifts under most-frequency present groups; underrepresented groups generally exhibit higher TVDs in the last layer compared to the intermediate layers. While intermediate layers also show this pattern, the spread across the TVDs of different groups is smaller. Since the last layer was found to be less effective for generalization in all the considered cases, sensitivity to distribution shifts provides a potential explanation for its limitations.

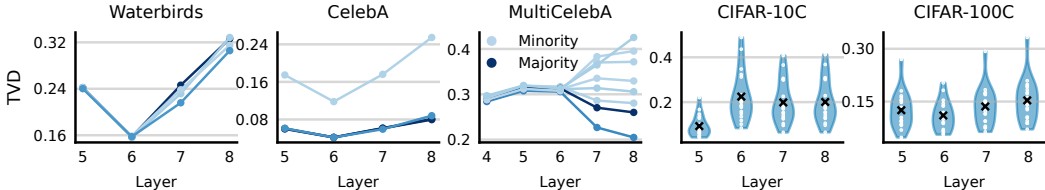

Figure 20: **Total variation distances (TVDs) between ID and OOD data.** We consider subpopulation shifts (three leftmost plots), where the lines are colored according to the fraction of samples the group is present in the training data, and input-level CIFAR shifts (two rightmost plots).

**Layer sensitivity predicts performance.** For the input-level shifts, features across layers for CIFAR-10C and CIFAR-100C exhibit similar TVDs, but the TVDs are generally lower for the intermediate layers compared to the last layer. While we have shown in the previous sections that the last layer is less effective for generalization, we additionally show that higher TVD values are associated with lower performance (Fig. 21).

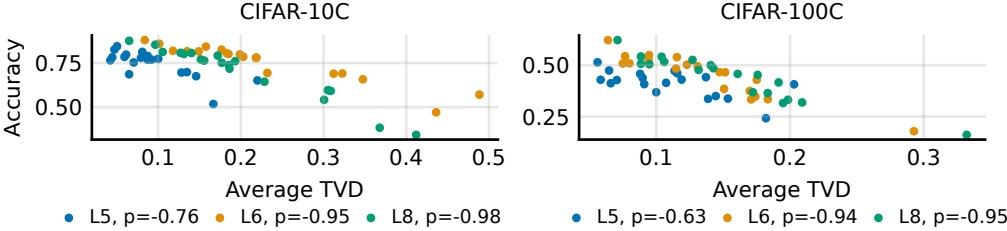

Figure 21: **Layer sensitivity.** More sensitive layers are associated with lower performance on CIFAR-10C and CIFAR-100C datasets. Layer 7 is excluded for CIFAR-10C due to it being near-identical to layer 8.

## C ADDITIONAL RESULTS

### C.1 RELATION TO NEURAL COLLAPSE

We discuss related work by (Li et al., 2022) and additionally inspect the relations between OOD generalization and Neural Collapse (NC) (Papyan et al., 2020). Neural Collapse describes a phenomenon where class means align along equiangular directions, and features collapse within their respective class means in the last layer of deep networks. This behavior is often quantified using metrics such as $NC_1$ (Papyan et al., 2020), which measures the alignment of within-class and between-class covariance matrices.

**Prelininaries.** Following (Papyan et al., 2020), $NC_1$ is defined as:

$$NC_1 := \frac{1}{K}\text{trace}\left(\mathbf{\Sigma}_W \mathbf{\Sigma}_B^\dagger\right), \tag{3}$$

where $\mathbf{\Sigma}_W \in \mathbb{R}^{d \times d}$ and $\mathbf{\Sigma}_B \in \mathbb{R}^{d \times d}$ are the within-class and between-class covariance matrices, respectively:

$$\mathbf{\Sigma}_W = \frac{1}{nK}\sum_{k=1}^{K}\sum_{i=1}^{n_k}(\mathbf{r}_i^{(k)} - \bar{\mathbf{r}}^{(k)})(\mathbf{r}_i^{(k)} - \bar{\mathbf{r}}^{(k)})^\top, \tag{4}$$

$$\mathbf{\Sigma}_B = \frac{1}{K}\sum_{k=1}^{K}(\bar{\mathbf{r}}^{(k)} - \bar{\mathbf{r}}_G)(\bar{\mathbf{r}}^{(k)} - \bar{\mathbf{r}}_G)^\top. \tag{5}$$

Here, $K$ is the number of classes, $n_k$ is the number of samples in class $k$, $\mathbf{r}_i^{(k)}$ is the feature representation of sample $i$ in class $k$, $\bar{\mathbf{r}}^{(k)}$ is the mean feature vector for class $k$, and $\bar{\mathbf{r}}_G$ is the global mean of all features.

In its original definition, $NC_1$ is applied to penultimate-layer representations. To study NC across intermediate layers $l$, $\mathbf{\Sigma}_W$ and $\mathbf{\Sigma}_B$ can be adopted to compute these statistics layer-wise using $\mathbf{r}_l(\mathbf{x})$, the representation of input $\mathbf{x}$ at layer $l$.

To approximate $NC_1$ behavior layer-wise, particularly under OOD conditions, we use the Class-Distance Normalized Variance (CDNV) metric (Galanti et al., 2021). CDNV is a practical alternative that focuses on class separability and compactness. For a given layer $l$, CDNV is defined as:

$$V_l(\mathbf{R}_i, \mathbf{R}_j) = \frac{\widehat{\text{Var}}_l(\mathbf{R}_i) + \widehat{\text{Var}}_l(\mathbf{R}_j)}{2\|\hat{\mu}_l(\mathbf{R}_i) - \hat{\mu}_l(\mathbf{R}_j)\|_2^2}, \tag{6}$$

where:

$$\hat{\mu}_l(\mathbf{R}_i) = \frac{1}{n_i}\sum_{\mathbf{x} \in \mathbf{X}_i}\mathbf{r}_l(\mathbf{x}), \quad \widehat{\text{Var}}_l(\mathbf{R}_i) = \frac{1}{n_i}\sum_{\mathbf{x} \in \mathbf{X}_i}\|\mathbf{r}_l(\mathbf{x}) - \hat{\mu}_l(\mathbf{R}_i)\|^2. \tag{7}$$

Here, $\mathbf{r}_l(\mathbf{x})$ represents the representation of sample $\mathbf{x}$ at layer $l$, $\mathbf{R}_i$ denotes the set of representations for class $i$, and $\mathbf{X}_i$ is the set of input samples for class $i$. For each layer $l$, the final CDNV score is computed as the mean of $V_l(\mathbf{R}_i, \mathbf{R}_j)$ over all pairs of classes $i \neq j$.

**Setup.** We compute $V_l(\mathbf{R}_i, \mathbf{R}_j)$ across ResNet18 layers to characterize NC behavior under distribution shifts for CIFAR-10 and CIFAR-100 ResNets, evaluating on CIFAR-10C and CIFAR-100C, respectively.

We note that this setup differs from (Li et al., 2022), where models were trained on ImageNet-1K and evaluated on downstream datasets like CIFAR-10. In our case, models are evaluated on the same dataset as the training dataset but under distribution shifts of covariates.

**Results.** We present results in Fig. 22 and Fig. 23 for CIFAR-10C and CIFAR-100C, respectively. We also indicate NC1 scores for the validation set of CIFAR-10 to compare how models' representation collapse differs under distribution shifts. We note that in both cases, *the model is most collapsed at the penultimate layer (layer 8), even on the downstream datasets.* These results differ from those in (Li et al., 2022), which suggest that the layer where the model is most collapsed depends on the downstream tasks. Given that the experiments in (Li et al., 2022) were conducted on general

pre-trained models (as opposed to task-specific datasets in our study), we believe this to be the key reason for the observed difference in behavior.

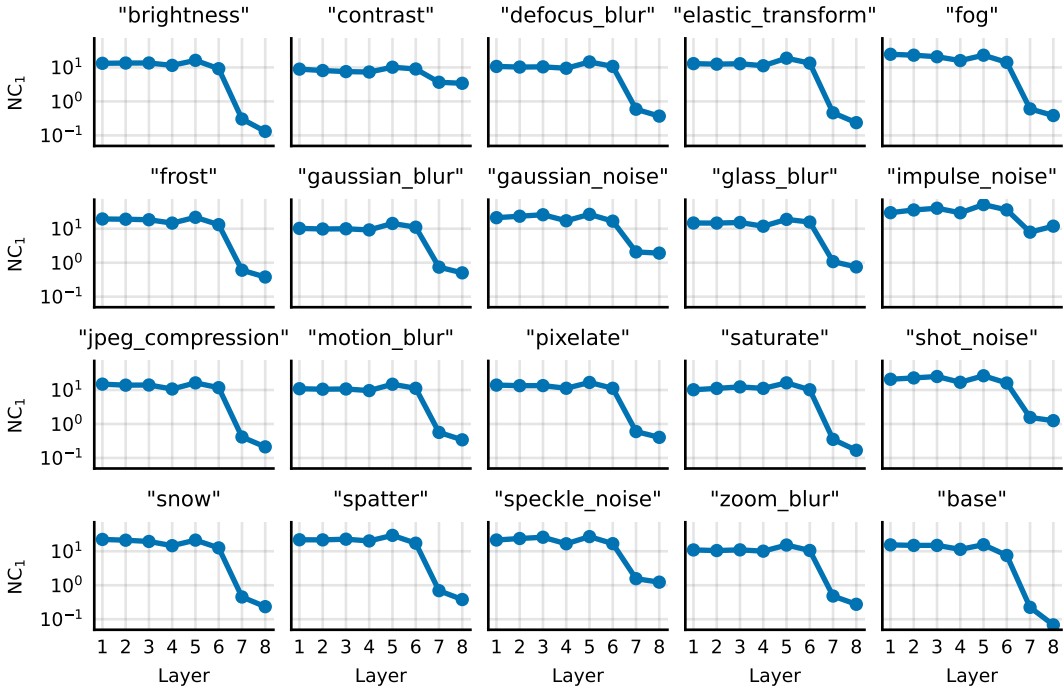

Figure 22: **Neural Collapse across layers using ResNet18 on CIFAR-10C datasets**.

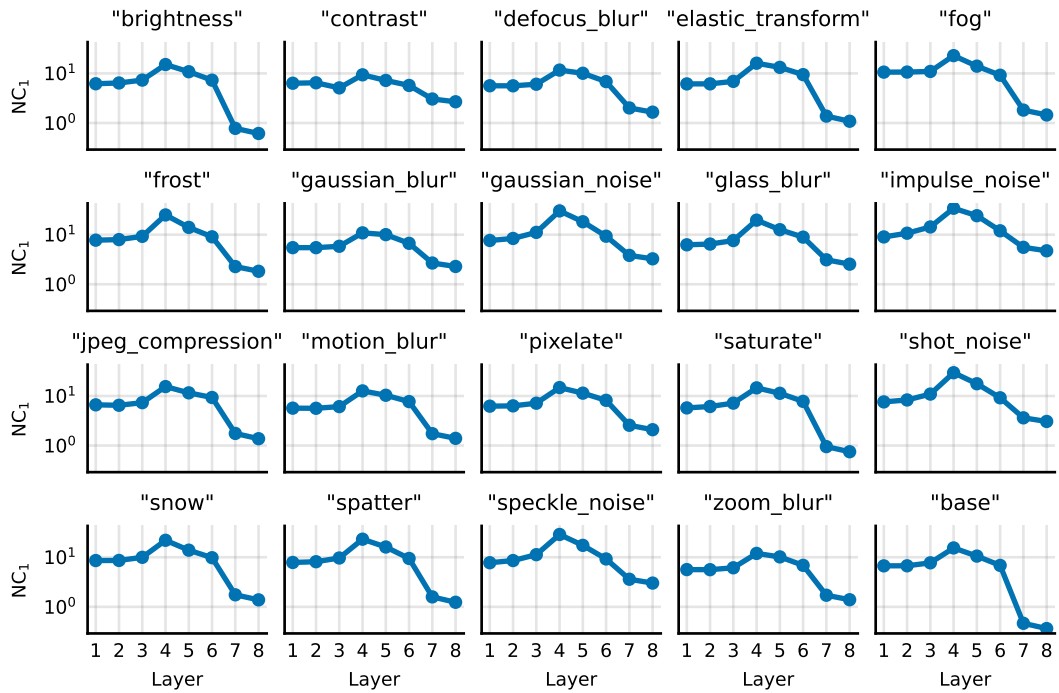

Figure 23: **Neural Collapse across layers using ResNet18 on CIFAR-100C datasets**.

## C.2 Non-linear ILCs

In this study, we have restricted intermediate-layer classifiers to linear probes to keep the complexity and computational costs low, ensuring the method remains practical. However, non-linear classifiers might better capture complex representations at intermediate layers, potentially improving performance. Moreover, the performance of non-linear probes can help assess whether the representations contain learnable information that is otherwise inaccessible to linear probes, providing further insights into the informativeness of these representations.

**Setup.** To investigate this, we train a one-hidden-layer MLP with 512 neurons and ReLU activation, which feeds into the linear classifier. We conduct few-shot experiments on CIFAR-10C and CIFAR-100C using ResNet18 and ViT. The setup and hyperparameter search follow the same protocol as in §4.2.1. We repeat the experiment with three random initializations of the classifier parameters to ensure robustness of the results.

**Results.** Non-linear probes improve performance on both CIFAR-10 and CIFAR-100, particularly in the third-to-last layer, as shown in Tables 5 and 6. This trend highlights the ability of non-linear probes to extract more complex patterns. For the last layer, the performance varies significantly across runs, with marginal to non-existent improvements on average. This variability suggests that, while some OOD-specific information might occasionally be accessible, the representations are generally insufficiently robust for OOD generalization and do not contain as rich information content as intermediate layers.

Table 5: **Results for ResNet18 on CIFAR-10 and CIFAR-100.**

| Layer ID | CIFAR-10 (%) | | CIFAR-100 (%) | |
|---|---|---|---|---|
| | **Linear Probes** | **Non-Linear Probes** | **Linear Probes** | **Non-Linear Probes** |
| $L-4$ | 74.57% | 75.07% | 30.40% | 34.10% |
| $L-3$ | 75.67% | 75.71% | 42.71% | 41.16% |
| $L-2$ | 70.75% | 71.27% | 45.01% | 45.83% |
| $L-1$ | 70.26% | 70.95% | 44.04% | 44.25% |

Table 6: **Results for ViT on CIFAR-10 and CIFAR-100.**

| Layer ID | CIFAR-10 (%) | | CIFAR-100 (%) | |
|---|---|---|---|---|
| | **Linear Probes** | **Non-Linear Probes** | **Linear Probes** | **Non-Linear Probes** |
| $L-4$ | 73.54% | 75.50% | 41.96% | 43.35% |
| $L-3$ | 75.19% | 77.12% | 41.81% | 43.31% |
| $L-2$ | 76.04% | 77.07% | 47.46% | 47.82% |
| $L-1$ | 74.71% | 74.48% | 47.69% | 47.47% |

## C.3 Impact of feature dimensionality on performance

In all of our experiments, we used features extracted at each depth of a neural network. For CNNs, this resulted in features whose dimensionality changed across layers, while for ViTs, the features had a fixed dimensionality across layers. In the case of CNNs, the feature dimensionality progressively decreased. For instance, for an input image of dimension $3 \times 32 \times 32$, as in CIFAR-10, ResNet18 produces features of dimensionality $8,192$ at layer 5, $4,096$ at layer 6, $2,048$ at layer 7, and $512$ at layer 8. In all of our experiments, linear probes were trained on the raw features (after flattening them), without applying any pooling techniques.

Given the large variation in feature dimensionality across layers, a potential concern is that the gains we observed in §4.2 and §4.3 arise from differences in feature dimensionality alone. We argue that this is not the case for two reasons. (1) For ILCs to succeed, the features in intermediate layers must be inherently informative of the task. If such features exist at these layers, dimensionality alone should not significantly influence performance (either the useful features are present or they are not). (2) Feature dimensionality does not explain the observed pattern in in-distribution (ID) performance,

where the highest performance is almost always achieved using last-layer features. As shown in §B.1.1, ID performance peaks, or is near the peak, when using penultimate layer representations in the classifier $\text{ILC}_{L-1}$ whose feature dimensionality is the smallest.

To validate this reasoning, we conduct an experiment to investigate how standardising feature dimensionality using PCA influences performance on CIFAR-10 with ResNet18.

**Setup.** Feature representations across layers were standardised using PCA. For each layer $i$, we extracted training representations $\mathbf{R}_i \in \mathbb{R}^{N \times d_i}$, where $N$ is the number of training samples and $d_i$ is the original feature dimensionality of the layer. These representations were centred as $\bar{\mathbf{R}}_i = \mathbf{R}_i - \text{mean}(\mathbf{R}_i)$, and the top 512 PCA directions $\mathbf{V}_i \in \mathbb{R}^{d_i \times 512}$ were computed. For test samples, representations $\mathbf{r}_i(\mathbf{x}) \in \mathbb{R}^{d_i}$ were centred using the training mean and projected onto the PCA basis as:

$$\mathbf{r}_i^{\text{PCA}}(\mathbf{x}) = (\mathbf{r}_i(\mathbf{x}) - \text{mean}(\mathbf{R}_i))\mathbf{V}_i \in \mathbb{R}^{512}.$$

This ensured that the dimensionality of all layer representations was fixed to 512. Linear probes were then trained and evaluated on these fixed-dimensional representations in a few-shot setup.

Table 7: **Few-shot accuracy (%) on CIFAR-10C with ResNet18 before and after PCA**. Dimensionality after PCA was fixed to 512 for all layers.

| Layer | Dim. (Before PCA) | Dim. (After PCA) | Acc. (Before PCA) | Acc. (After PCA) |
|---|---|---|---|---|
| 5 | $8,192$ | 512 | 74.3% | 71.4% |
| 6 | $4,096$ | 512 | 75.3% | 74.6% |
| 7 | $2,048$ | 512 | 70.4% | 71.3% |
| 8 | 512 | 512 | 69.9% | 69.9% |

**Results.** Table 7 shows the results. Performance trends remained consistent after PCA, with layer 6 achieving the highest accuracy in both setups. The minor differences in accuracy before and after PCA suggest that dimensionality alone does not drive the trends observed in layer-wise performance.

## C.4 LAYER-WISE TRANSFERABILITY ACROSS DISTRIBUTION SHIFTS

To better understand the role of individual layers in handling distribution shifts, we evaluate their cross-dataset and cross-shift performance while keeping the model architecture fixed. Specifically, we analyse the transferability of layers in ResNet18 by examining their performance on CIFAR-10C and CIFAR-100C under various corruptions in the few-shot setting. This experiment assesses how well features from a specific layer generalise across datasets with similar distribution shifts and measures the extent to which layers transfer across distribution shifts and datasets.

**Setup.** For each corruption type in CIFAR-10C, we evaluate the performance of layers $\ell \in \{3, \dots, 8\}$ when transferring to the same corruption type in CIFAR-100C. Linear probes are trained separately on features extracted from CIFAR-10C and CIFAR-100C, following the setup described in §4.2.1. The best probes are selected based on the validation split. For any given dataset, distribution shift, and layer, we compare how the probes perform on CIFAR-10C versus CIFAR-100C to evaluate transferability.

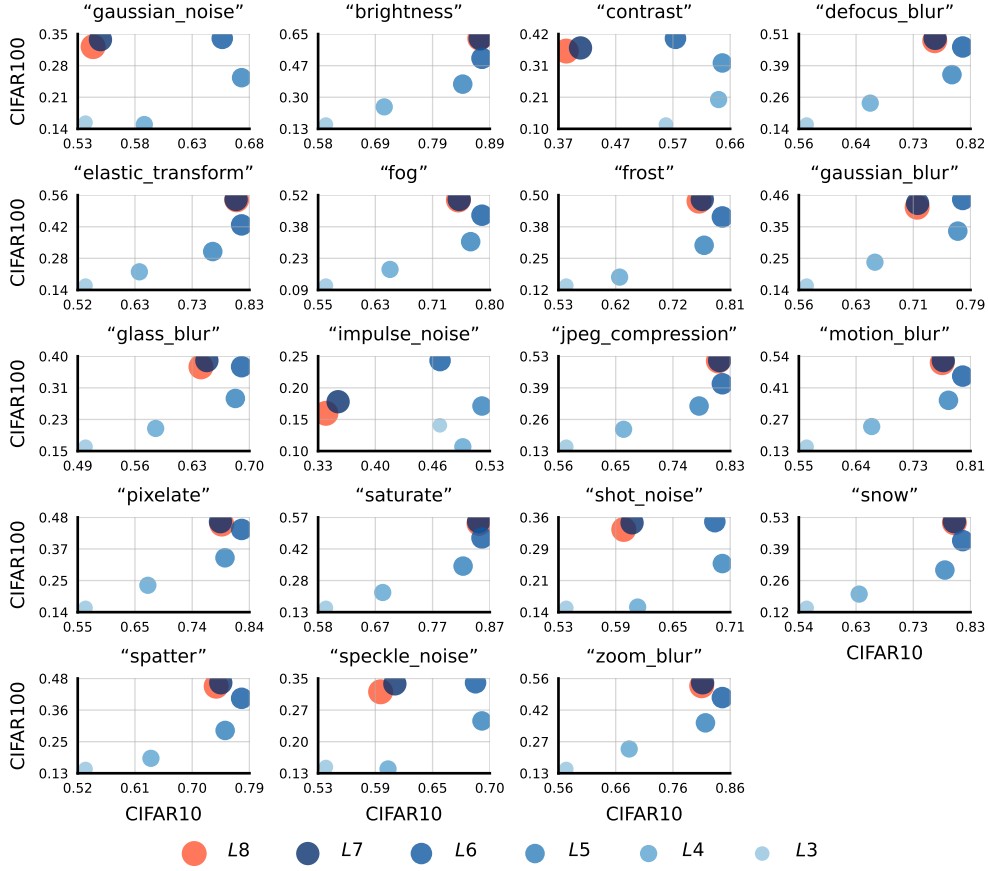

Figure 24: **Layer transferability between CIFAR-10C and CIFAR-100C on ResNet18**. Each scatter plot represents a specific corruption type, with the x-axis showing the accuracy on CIFAR-10C and the y-axis showing the accuracy on CIFAR-100C. Points denote the performance of individual layers $\ell \in \{3, \ldots, 8\}$.

**Results.** The results are summarised in Figure 24. We highlight two key observations regarding layer transferability across datasets and distribution shifts:

(1) **Pen-penultimate layer consistently outperforms penultimate layer.** For each corruption, the pen-penultimate layer achieves higher performance than the penultimate layer on both CIFAR-10C and CIFAR-100C.

(2) **Dataset-specific optimal layers.** The layer achieving the highest performance on CIFAR-10C is often not the best-performing layer on CIFAR-100C for the same corruption. This suggests that while layers transfer reasonably well, certain layers may provide features that are better suited for a specific dataset, even under similar distribution shifts.

This analysis shows that transferable information is distributed non-uniformly across layers and varies by dataset, even with the same architecture, highlighting the importance of fine-grained layer evaluations under distribution shifts.

