# OpenReview forum: "Intermediate Layer Classifiers for OOD generalization"
_ICLR.cc/2025/Conference — ICLR 2025 Poster_

### Official Review · Reviewer_P4oy · 2024-10-27

**Soundness:** 3
**Presentation:** 3
**Contribution:** 2
**Rating:** 6
**Confidence:** 3

**Summary:**

This paper demonstrates that leveraging intermediate features outperforms conventional approaches relying on last-layer features for OOD generalization. The claim is substantiated through extensive experiments across diverse datasets, varying data quantities, model architectures, and types of distribution shift. Additional analysis on feature sensitivity further reveals that intermediate features exhibit greater robustness to distribution shifts.

**Strengths:**

* The paper is well-written and clearly presented.
* The experiments are comprehensive, covering multiple aspects.
* The analysis of feature sensitivity is interesting.

**Weaknesses:**

* The most convincing experimental results supporting the paper’s claim are observed on CNNs, whereas the performance gains on (currently) more widely used transformer-based architectures are less pronounced, with intermediate features providing only marginal improvement over last-layer features. It may worth discussing the potential reason for this performance disparity.
* More critically, while the paper offers detailed descriptions of the data and model setups, it lacks a clear feature extraction protocol. For instance, when using an intermediate feature with dimensions $256 \times 4 \times 4$, do the authors apply pooling before probing or directly flatten the representation? This question is important since ConvNets almost always have a feature dimension change across layers and this may contribute to the performance difference between intermediate features and last-layer features.

**Questions:**

Please see the Weaknesses.

---

> ### Author Response · Authors · 2024-11-20
>
> Thank you for insighftul comments. We address them below.
>
> **W1: Gap between transformers and CNNs**
>
> Thank you for raising this point. We currently lack a definitive explanation but hypothesize it may relate to differences in receptive fields or architectural biases between CNNs and transformers. We will add a discussion on this in the paper
>
> **W2: Influence of Feature Dimensionality**
>
> We acknowledge that the feature extraction protocol was insufficiently clarified in the current draft and will be addressed this in the revised manuscript.
>
> Specifically, in this work, we did not apply any form of feature pooling before probing. While we understand the reviewer’s concern regarding the potential impact of feature dimensionality on performance, we believe this factor is secondary to the informativeness of features themselves. Our perspective is twofold:
> 1. For ILCs to succeed, the features in intermediate layers must be inherently informative of the task. If such features exist at these layers, dimensionality alone should not significantly influence performance (either the useful features are there or they are not).
> 2. Feature dimensionality does not explain the observed pattern in in-distribution (ID) performance, where the highest performance is almost always achieved using last-layer features. As shown in Figure 18 of the Appendix, ID performance peaks, or is near the peak, when using penultimate layer representations in the classifier $ILC_{L-1}$.
>
> To ensure dimensionality is not a confounding factor, we conducted an additional experiment on CIFAR-10C using ResNet18. Initial output dimensions of layers were as follows: Layer 4: $16,384$, Layer 5: $8,192$, Layer 6: $4,096$, Layer 7: $2,048$, and Layer 8: $512$.
>
> Representations $\mathbf{R}_i$ were extracted from training samples at each layer $i$, centered as $\bar{\mathbf{R}}_i = \mathbf{R}_i - \text{mean}(\mathbf{R}_i)$, and the top $512$ PCA directions $\mathbf{V}_i \in \mathbb{R}^{d_i \times 512}$ were computed. For a test $\mathbf{x}$, its representation $\mathbf{r}_i(\mathbf{x})$ were centered using the training mean and projected as $(\mathbf{r}_i(\mathbf{x})- \text{mean}(\mathbf{R}_i)) \mathbf{V}_i$. Linear probes were trained and evaluated on these projections in few-shot setup. This setup ensures fixed-dimensional representations across layers, isolating the role of informativeness. We present the results in the table below.
>
> | Layer | Accuracy (Before PCA) | Accuracy (After PCA) |
> |-------|-----------------------------|----------------------------|
> | 5     | 74.3%                      | 71.4%                     |
> | 6     | 75.3%                      | 74.6%                     |
> | 7     | 70.4%                      | 71.3%                     |
> | 8     | 69.9%                      | 69.9%                      |
>
> Compared to the results before PCA, the mean accuracy differences are minor, particularly for Layer 6 (the highest-performing layer in both setups). This supports the conclusion that dimensionality alone does not explain the trends observed in layer-wise performance, as the fixed-dimensionality setup after PCA produces similar relative patterns across layers.
>
> We're happy to engage in further discussion.

---

> > ### Comment · Reviewer_P4oy · 2024-11-22
> >
> > I appreciate the additional experimental results provided by the authors.
> >
> > I think this PCA experiment is highly persuasive and significant. I recommend that the authors consider incorporating it into the main body of the paper in a future updated version.
> >
> > As my primary concern has been addressed, I am increasing my score from 5 to 6.

---

> > > ### Author Response · Authors · 2024-11-24
> > >
> > > We thank the reviewer for the score increase. We have updated the PDF to include a reference in the main text to the results on the influence of feature dimensionality in Appendix C.3. We sincerely appreciate the suggestion.

---

### Official Review · Reviewer_Nh8y · 2024-11-03

**Soundness:** 3
**Presentation:** 3
**Contribution:** 3
**Rating:** 6
**Confidence:** 3

**Summary:**

This paper questions the use of last-layer representations for out-of-distribution (OOD) generalization due to their sensitivity to data distribution shifts. Instead, by introducing intermediate layer classifiers (ILC), the paper shows that intermediate layer representations often outperform penultimate layer representations in generalization.

**Strengths:**

The paper is well-written and easy to follow. The observation is interesting that intermediate layers may perform better than the last layer in generalization. And, the paper introduces a new metric named "sensitivity score" to measure the sensitivity of each layer to distribution shifts.

**Weaknesses:**

1. The paper discusses a concept called "information content", but it lacks some detailed explanation. The authors need to demonstrate why the accuracy on OOD tasks can characterize "information content".
2. The potential impact is not so clear. The observation is interesting, but it may be difficult to find the "best" layer in practice.
3. It is better to provide more theoretical backups to support the experimental findings.

Small typos: "atop" (line 469); "in earlier layers" (line 497); "$\pi$" not defined (line 500)

**Questions:**

1. Could the authors provide more explanations about the feasibility of using the sensitivity score as defined in the paper? Why is this score appropriate for characterizing the sensitivity?
2. All the experiments are based on the task of image classification. What will the results become when dealing with other tasks besides image classification?

---

> ### Author Response · Authors · 2024-11-20
>
> Thank you for insighftul comments. We address them below.
>
> **W1: Usage of "information content": relationship between accuracy and "information content".**
>
> Thank you for raising this point. We refer to “information content” as the representations’ informativeness about the task under distribution shifts. Accuracy is used as a proxy for this informativeness, as it directly reflects how well the features support the task. We focus on the lack of informativeness in the penultimate layer because its representations are optimized for the original task, achieving high accuracy in-distribution. As shown in Section 4.2.1 (*Information Content for OOD Generalization at Last Versus Intermediate Layer*), these representations often perform worse on OOD tasks compared to intermediate-layer probes, highlighting their limited robustness for OOD generalization and thus their lack of information content, even with abundant OOD data.
>
> To further support this argument, we conducted additional experiments using non-linear probes (e.g., multi-layer perceptrons) on intermediate-layer representations, as described in Appendix C.2 (in the updated version of the PDF). The results showed negligible performance improvements compared to linear probes. *This suggests that the lack of performance is not due to the probe’s limited capacity but rather the absence of learnable information in the representations*.
>
> **W2: It may be difficult to find the "best" layer in practice**
>
> We agree that finding the best layer is not always straightforward. However, model selection typically involves a held-out testing set, which can include OOD datapoints [a, b, c]. Under this setting, selecting the best layer is analogous to selecting the best hyperparameters or training epochs when using a last-layer classifier. Since we assume access to already-pretrained models, this approach only requires training a set of linear probes, making it computationally similar to last-layer retraining methods [a].
>
> [a] Kirichenko, Polina, Pavel Izmailov, and Andrew Gordon Wilson. "Last Layer Re-Training is Sufficient for Robustness to Spurious Correlations." ICLR 2023 (2023).
> [b] Sagawa, Shiori, et al. "Distributionally robust neural networks for group shifts: On the importance of regularization for worst-case generalization." arXiv preprint arXiv:1911.08731 (2019).
> [c] Gulrajani, Ishaan, and David Lopez-Paz. "In search of lost domain generalization." arXiv preprint arXiv:2007.01434 (2020).
>
> **W3: Lacking theoretical backing**
>
> We agree that a mathematical understanding of this phenomenon would be valuable. However, we believe this is challenging, as it requires developing insights specific to DNNs that extend beyond analyses of simpler, two-layer networks. Even for last-layer retraining methods, theoretical understanding is limited. Our empirical findings suggest that intermediate layers capture more general, task-invariant information and handle OOD data more similarly to ID data compared to the penultimate layer. Understanding how features evolve across layers, particularly why the penultimate layer sometimes also captures task-invariant features, remains an open question and is largely addressed in empirical studies at present.
>
> **Q1: Feasibility and appropriateness of feasibility score**
>
> The key idea is to measure how features change from ID to OOD data. This is done per group (or per class, in classification problems), accounting for how each sub-cluster within the group spreads under distribution shifts. The score reflects a layer’s ability to preserve features consistent across ID and OOD data. By normalising with within-group distances, the metric avoids bias from the variability of ID data within a layer's features. Additionally, the score aligns with observed layer behavior, where intermediate layers exhibit lower sensitivity and greater robustness, supporting its validity as a meaningful measure of sensitivity. For these reasons, we believe this score is appropriate for evaluating feature sensitivity.
>
> **Q2: Tasks other than image classification**
>
> While we don’t have results for tasks beyond image classification, we hypothesize that the outcomes depend on the nature of the supervision signal. Tasks with fine-grained supervision may benefit less from intermediate layers, whereas tasks with coarser supervision signals may rely on them more.

---

> > ### Comment · Reviewer_Nh8y · 2024-11-24
> >
> > Thanks to the authors for their response, which addresses most of my concerns. I still recommend acceptance and maintain my rating.

---

### Official Review · Reviewer_G9DE · 2024-11-04

**Soundness:** 3
**Presentation:** 3
**Contribution:** 3
**Rating:** 6
**Confidence:** 4

**Summary:**

The paper investigates how intermediate layers in deep neural networks (DNNs) can improve out-of-distribution (OOD) generalization, challenging the common practice of relying solely on the last layer's representations. The authors use the notion of Intermediate Layer Classifiers, which consist of linear probes attached to each intermediate layer of the network but trained specifically to perform OOD classification. Two scenarios as considered: one "few-shot", in which few OOD samples are available for training, and one "zero-shot", where no OOD samples are available, under several types of distribution shifts, datasets, and architectures. They conclude that in both cases training classifiers on intermediate layer representations tends to perform better than last-layer retraining, and suggest that one of the reasons for this phenomenon is that intermediate representations are less sensitive to distribution shifts.

**Strengths:**

- The paper is very well-written.
- The paper assesses intermediate-layer performance across a variety of tasks, datasets, and architectures, which strengthens the generalizability of the findings.
- Interesting discussion on why intermediate layers might generalize better to OOD data, including their reduced sensitivity to distribution shifts and improved data efficiency.
- By demonstrating that intermediate layers can achieve competitive results even in few-shot or zero-shot settings, this work provides practical benefits, especially for applications where OOD data is scarce or unavailable.

**Weaknesses:**

__Sensitivity Analysis:__ although the sensitivity analysis aims at showing the stability of intermediate-layer features, it lacks sufficient depth on why some intermediate layers are more stable than others in different types of shifts, particularly for different architectures. The inclusion of a summary of any trends observed across architectures/datasets/shifts would be helpful.

__Figure 10:__ The point made in the description of Figure 10 is not clear from the plots shown, especially since it is not clear how many points occlude one another in the scatter plots. What dataset was used here, and what model? The quantitative analysis is also confusing: although the authors say that an increasing separation exists in earlier layers, I believe this only holds for the minority groups. Perhaps the authors should clarify this.

__Terminology:__ I found the terminology a bit confusing when dealing with training points vs. testing points vs. probe points vs. OOD points. Namely, I had trouble following which sets contained OOD points vs. ID points, especially in the zero-shot section. It would be advisable to always point out which points are OOD.

__Novelty:__  The literature reviewed for related work seems superficial. A notable example of a paper missing from the references is:

[1] Yosinski, J., Clune, J., Bengio, Y., & Lipson, H. (2014). "How transferable are features in deep neural networks?"_Advances in Neural Information Processing Systems_, 27. https://proceedings.neurips.cc/paper_files/paper/2014/hash/375c71349b295fbe2dcdca9206f20a06-Abstract.html

which explored a method for quantifying the transferability of features from each layer of a deep network, including their applicability to shifts in the data (e.g., different subsets, and even different classes).

Furthermore, the novelty of this contribution is somewhat diminished given that an earlier paper (accepted at last year's ICLR) has also proposed evaluating intermediate layers’ effectiveness in generalization for OOD samples:

[2] Gerritz et al. (2024) "Zero-shot generalization across architectures for visual classification." _The Second Tiny Papers Track at ICLR 2024_, https://openreview.net/forum?id=orYMrUv7eu

In fact, an extended version of that paper (from the same group) also used linear probes to quantify the out-of-sample accuracy of intermediate representations:

[3] Dyballa et al. (2024). A separability-based approach to quantifying generalization: which layer is best?. _arXiv preprint arXiv:2405.01524_, https://arxiv.org/abs/2405.01524

This paper would certainly benefit from acknowledging that similar observations have been previously made when performing category discovery on unseen classes (as opposed to the distributional shifts here studied).

__Frozen weights:__ The fact that the authors only considered frozen, pre-trained weights in all scenarios studied is somewhat unrealistic or at least incomplete. In any practical application of this method, if it is feasible to fine-tune the pretrained model on one's own data, then that will certainly be preferable. See reference [1] for an example in which both cases (frozen weights vs fine-tuned) are considered. Specifically for this paper, it would be interesting to verify whether the comparisons between penultimate layer and intermediate layer outputs hold in that scenario.

**Questions:**

- __Figure 3:__ which ResNet was used to produce the bar plot?
- __Figure 4:__ which ResNets were used to produce these plots? Was any trend observed when comparing the different ResNet sizes?
- __Section 4.3:__ can we really call "zero-shot" a setting in which the "OOD" data is composed of ID samples with added noise? That is a standard modification used in data augmentation (used to increase the number of ID samples), for example, so such a "distribution shift" seems extremely mild to be considered zero-shot. I would have liked to see a discussion of the different shifts chosen compare to one another in terms of difficulty.
- Did the authors find that any particular architecture had intermediate representations consistently outperforming last-layer retraining?
- Did the best layer for a particular dataset seem to agree with the best layer for other datasets, for the same model?

---

> ### Author Response · Authors · 2024-11-20
> **(1/2)**
>
> Thank you for your insightful review.
>
> **W1: Sensitivity analysis lacks depth on why some intermediate layers are more stable than others in different types of shifts**
>
> Thank you for the suggestion. We agree that understanding why some intermediate layers are more stable than others is an important question, especially across architectures. Answering this is challenging, as even within a single dataset like CIFAR-10C, performance varies significantly across different types of noise (see Figure 15 in the Appendix). This suggests that the stability of intermediate layers depends on both the nature of the shift and the architecture.
>
> We are actively analyzing these trends across architectures and will incorporate the findings into the paper.
>
> **W2: Analysis in Fig. 9 / Fig. 10 isn't clear**
>
> Thank you for the feedback. We clarify the intended points and outline the changes made to the paper for better clarity.
>
> *Quantitative Analysis (Figure 9)*
> We argue that earlier layers exhibit decreasing separation or sensitivity, particularly for minority data, which we regard as OOD. This aligns with your observation that the trend primarily holds for minority groups. The same conclusion extends to other shifts, such as CIFAR-10C and CIFAR-100C, as shown in Figure 19 in the appendix. The key point is that for the minority group, the penultimate layer separates ID and OOD points most distinctly.
>
> *Qualitative Analysis (Figure 10)*
> The plot shows train and test samples under PCA projections using the MultiCelebA dataset and a ResNet18 model. Thank you for noticing this missing detail. We have included this information in the updated version of the paper. We have also clarified the penultimate layer's separation for a minority group in the caption.
>
> **W3: Terminology: training points vs. testing points vs. probe points vs. OOD points**
>
> To clarify this, we have explicitly indicated in the few- and zero-shot sections whether $\mathcal{D}_{\text{probe}}$ is sampled from ID or OOD distributions. Additionally, we have referenced Section 3.1 in these sections and included a graphical representation of the data used in the probes (see Table 1).
>
> **W4: Relation to previous works**
>
> Thank you for these references. The key distinction is that the cited works primarily focus on generalization to novel class splits, whereas our work addresses scenarios where the task and visual variations remain closely aligned with the original distribution. We have updated the manuscript to relate our work to the previous studies.
>
> **W5: Authors only considered frozen, pre-trained weights in all scenarios studied, which is somewhat unrealistic or incomplete**
>
> We agree that fine-tuning pre-trained models is common in practice. However, all models in our experiments were tailored to their respective datasets. For example, CelebA and Waterbirds models were fine-tuned on their specific datasets starting from ImageNet-pretrained weights, while CIFAR-10 models were trained from scratch. This setup ensured that each model was well-aligned with its target dataset, enabling a consistent evaluation of intermediate-layer representations without introducing additional variability from further fine-tuning.

---

> ### Author Response · Authors · 2024-11-20
> **(2/2)**
>
> **Q1: Which ResNets were used to produce the plots (Figures 3, 4)?**
>
> Figure 3: ResNet18. Figure 4: ResNet-50 (Waterbirds, CelebA) and ResNet18 (remaining plots). This information has been added to the updated manuscript, thank you for pointing this out.
>
> **Q2: Was any trend observed when comparing different ResNet sizes in Figure 4?**
>
> The influence of ResNet size appears minimal. In cases with limited training data for probes, the gap between ILCs and last-layer retraining methods depends more on the dataset than the architecture.
>
> **Q3: Can we really call "zero-shot" a setting in which the "OOD" data is composed of ID samples with added noise?**
>
> Thank you for this perspective. To clarify, in all "zero-shot" settings, we do not use any OOD data during training (e.g., no noise is added to ID samples during this phase). While we agree that noise addition conceptually represents a mild distribution shift, we note the following:
> 1. In CIFAR-10C/-100C experiments, we used the highest noise level (level 5).
> 2. Classifiers trained on noise-free CIFAR-10/100 datasets perform poorly on noisy datasets, with a performance drop of ~25% on CIFAR-10C (see Figure 1), illustrating the practical significance of this shift.
>
> Given that the model was not originally trained on all noise variations and exhibits a drastic performance drop on noisy datasets, we believe this constitutes a valid OOD task.
>
> **Q4: Did the authors find that any particular architecture had intermediate representations consistently outperforming last-layer retraining?**
>
> Yes, intermediate representations consistently outperformed last-layer retraining across nearly all dataset-architecture pairs. ResNets, in particular, exhibited consistent trends in all experiments. The only exceptions were CelebA and Waterbirds, where intermediate representations performed similarly to last-layer retraining in the few-shot setting with the full OOD dataset used for training. However, in zero-shot settings and cases where only a fraction of the data was used, ILCs continued to outperform last-layer retraining. We hope this answers your question, and we are happy to provide further clarification if needed.
>
> **Q5: Did the best layer for a particular dataset seem to agree with the best layer for other datasets, for the same model?**
>
> Thank you for this interesting question. Figure 18 in the Appendix shows accuracies per layer in both zero- and few-shot settings. While the exact position of the best layer varies across datasets, it is typically located in the second half of the network, as discussed in Section 5.

---

> ### Author Response · Authors · 2024-11-24
>
> **W1: Trends across architectures/datasets/shifts; Q5: Agreement of layers across shifts for the same architecture**
>
> We have extended our analysis to provide a fine-grained view of the alignment between ILCs performance across layers, datasets, and shifts for CIFAR-10C and CIFAR-100C. Detailed results are included in Appendix C.4 of the updated PDF.
>
> Across all tested shifts on CIFAR-10C and CIFAR-100C for ResNet18, the intermediate layer ($\ell = 7$) _always_ outperforms the penultimate layer ($\ell = 8$). Regarding agreement across datasets for the same shift, the best-performing layer on CIFAR-10C is often not the best-performing layer on CIFAR-100C.
>
> Should additional questions regarding cross-transfer among shifts or datasets arise, we would be happy to engage in further discussion.

---

> ### Comment · Reviewer_G9DE · 2024-11-24
>
> I thank the authors for their detailed response. I believe the paper has significantly improved after the authors addressed my concerns and those from other reviewers, so I am increasing my score to 6.

---

### Official Review · Reviewer_ncZg · 2024-11-05

**Soundness:** 3
**Presentation:** 3
**Contribution:** 3
**Rating:** 8
**Confidence:** 3

**Summary:**

This paper shows that in ResNets and ViTs, intermediate layers classification is more robust to the distribution shifts of the out-of-distribution samples (OODs). In particular it shows that the features of the pen-pen-ultimate layer consistently outperform the features of the pen-ultimate layer, when it comes to the zero-shot and few shot cases. These results are interesting, as they contradict the current belief that it is enough to retrain the last layer, that is the classification layer. It is assumed that each layer computes interesting invariants of an input image, and this paper raises the possibility that some of these invariants get lost in downstream layers.

**Strengths:**

This is a very intriguing paper, that raises the question of what happens in the last layers, that overcomes generalization to OOD samples?

**Weaknesses:**

While analyzing the depth and the sensitivity as important factors in generalization ability, it would be interesting to understand what happens in the last layers that reduces generalizability?

**Questions:**

In principle, the features of intermediate layers, are propagated down all the way to the pen-ultimate layer, through skip connections, in both ResNets and ViTs. However, they are added before every layer with the output of the previous layer.

Is this the reason of why generalization ability decreases?

From the point of view of back-propagation the addition does not matter, as the derivative of a sum is the sum of the derivatives. However, this addition may clutter the features?

---

> ### Author Response · Authors · 2024-11-20
>
> Thank you for an insightful review.
>
> **W1: It would be interesting to understand what happens in the last layers that reduces generalizability?**
>
> We agree that understanding this is important. Our findings suggest that the penultimate layer’s features are highly specialised for in-distribution tasks. This specialisation may reduce robustness to distribution shifts. The sensitivity analysis in section 5.2 supports this, showing that penultimate-layer features are more sensitive to shifts than those in intermediate layers. This remains an open question, and we would be interested in exploring it further.
>
> **Q1: Is the reason for the decrease in generalization due to skip-connections?**
>
> Thank you for raising this point. Skip connections propagate intermediate features through the network. In theory, this should preserve generalisable information. However, our results suggest that the penultimate layer aggregates features in a way that does not fully retain robustness. Figure 6 shows GoogLeNet performance, which does not use residual connections like ResNets but still exhibits a large gap in performance between intermediate and penultimate layers. This indicates that the observed behaviour is not solely due to residual connections. We do not have conclusive evidence linking skip connections to reduced generalisability, but this hypothesis is interesting and merits further investigation.

---

### Official Review · Reviewer_TgHf · 2024-11-07

**Soundness:** 3
**Presentation:** 3
**Contribution:** 3
**Rating:** 6
**Confidence:** 4

**Summary:**

The authors explore whether the intermediate layers offer better out-of-distribution (OOD) generalisation. They found that in a number of settings, including zero- and few-shot, they do and demonstrate the limits of penultimate layer representation utility. Most interestingly, the authors claim that ‘earlier-layer representations without fine-tuning on the target distribution often fare competitively with the last-layer retraining on the target distribution’.
The contribution of the paper is empirical, and the authors support the claims with extensive number of experiments.

**Strengths:**

Pros:
- Significance: The question of generalisation, especially to the zero-shot scenarios, is an important one. While I think it’s been tremendous work done, s I discuss in comment 2, it is important to discuss a number of limitations such as selection of evaluation dataset,
- Correctness: I haven’t spotted any incorrect statements in the paper
- Motivation: the paper challenges the conventional wisdom of using the last layer for zero and few-shot learning, and I believe it’s a great motivation
- Reproducibility: I’ve checked the experimental setting and it looks reproducible to me. I would also highlight that the description of the experiments is exceptionally clear
- Clarity: the paper is very clearly written (apart from a couple of comments below)

**Weaknesses:**

Cons:
- Clarity and limitations: a few questions on the experimental setting

**Questions:**

1. “Line 198-199: using  data that can be either ID or OOD, depending on whether  the setup is zero-shot or few-shot.” Not sure I get this phrase, why would the data be selected ID for the zero-shot scenario and OOD for the few-shot scenario? Shouldn’t it be both ID and OOD for both cases (or just OOD)?
2. The biggest question and concern is the inclusion criteria for both datasets and the models. The dataset performance beyond most-known datasets such as CIFAR-10 and CIFAR-100 may differ not least due to the fine-grained nature of some of these datasets. Figure 7 from Tobaben et al (2023), for example, chose to include VTAB-1k datasets to test the performance on the different datasets. On the inclusion of the models, on one hand, it seems to give a fair share of ViTs and ResNets, however it would be great to spell out the inclusion criteria: why did the authors choose this particular selection of the model? Separate subsection in  Section 4 seems to be an appropriate place for this.
3. The authors can argue that it’s not within the scope, so it would be a fair play, however, it would be interesting to see the following. It appears that the models have been trained on the clear target dataset (i.e., the authors  evaluate the OOD performance of the CIFAR-100 trained model), and I wonder if they tried whether these findings generalise for models, pre-trained on generic large-scale data (i.e. ImageNet-pretrained or pretrained models such as DinoV2 or other varieties of pretrained ViT)? This may be useful in finding out whether the conclusion of the paper remain valid for the scenarios when

Tobaben et al (2023) On the Efficacy of Differentially Private Few-shot Image Classification, TMLR 2023

---

> ### Author Response · Authors · 2024-11-20
>
> Thank you for insightful comments.
>
> **Q1: Why would the data be selected ID for the zero-shot scenario and OOD for the few-shot scenario?**
>
> In the *zero-shot scenario*, $D_\text{probe} \sim P_\text{ID}$, as no OOD data is used for training the probes—only the original training data that the backbone was trained on. In the *few-shot scenario*, $D_\text{probe} \sim P_\text{OOD}$, aligning with the last-layer retraining paradigm, where limited OOD samples are used for adaptation.
>
> We’d be happy to discuss this further if we misunderstood your question.
>
> **Q2: Inclusion criteria of models and datasets isn't clear**
>
> Thank you for the question. We agree that the inclusion criteria should have been explained in more detail. The selection of models and datasets is closely intertwined, as we wanted to avoid training models on ID datasets ourselves. Each (model, dataset) pair typically requires specific recipes, and we aimed to prevent misrepresenting results by relying exclusively on publicly available pre-trained models. Essentially, we focused on distribution shifts where neural networks tend to underperform and selected datasets within each domain based on their popularity and usage in the community.
>
> We deliberately excluded VTAB-1k datasets because they consist of subsets from multiple datasets, conflicting with our focus on evaluating models trained within a single domain. VTAB-1k evaluates performance across diverse datasets, making it more suitable for studying generalization across tasks rather than distribution shifts within a domain. This fundamental difference in scope is why we did not include VTAB-1k in our evaluation.
>
> We have updated Section 4.1 to include the selection criteria for primarily using ViTs and ResNets due to their differing inductive biases under "DNN model usage and selection" and the dataset selection criteria in Table 2. Due to space constraints, adding a new subsection was challenging, so these updates were integrated into existing sections.
>
> **Q3: ILCs on generic pre-trained models on dataset shifts**
>
> Thank you for raising this interesting question. Our work closely follows the last-layer retraining literature, and we deliberately restricted the scope to distribution shifts within a dataset, rather than transfer across datasets.
>
> Testing ILCs on generic pre-trained models like ImageNet-pretrained ViTs or DinoV2 would not align with our zero-shot framework, a key contribution of our work, where no OOD data is used for training. These models inherently involve pretraining on diverse datasets, which conflicts with the assumptions of our setup. While testing such models would be interesting, it falls outside the intended scope of our current study.

---

> > ### Comment · Reviewer_TgHf · 2024-11-21
> >
> > Dear authors,
> >
> > Many thanks for the thorough rebuttal, it addresses my concerns. I am also checking the rest of the reviews and will come back on it soon.
> >
> > "selected ID for the zero-shot scenario and OOD for the few-shot scenario"
> > Would be great if the authors amend the text of the paper accordingly.

---

> ### Author Response · Authors · 2024-11-21
>
> Dear reviewer TgHf,
>
> Thank you for the comment. We amended the text to reflect your suggestion and additionally explicitly indicated the distributions from which the dataset is drawn in both scenarios. We would be happy to engage in further discussion if any questions arise.

---

> ### Comment · Reviewer_TgHf · 2024-11-22
>
> I've checked the comments by the other reviewers, and I think I still recommend acceptance. Amongst the concerns, I think it is important that the authors put the answer to the question about feature extraction protocol from Reviewer P4oy into the paper.

---

> > ### Author Response · Authors · 2024-11-24
> >
> > Thank you for your thorough review. We have amended the text to reflect your suggestion and have included the reference to the feature extraction protocol question raised by Reviewer P4oy in the updated paper.

---

### Official Review · Reviewer_JuAB · 2024-11-08

**Soundness:** 3
**Presentation:** 4
**Contribution:** 2
**Rating:** 6
**Confidence:** 3

**Summary:**

This paper investigates the use of intermediate layer linear probing classifiers to enhance a model's performance on out-of-distribution (OOD) datasets, where the test data distribution differs significantly from the training data distribution. People usually assume that the network can act as a combination of general feature extraction(early layers) plus in-domain task specialization(later layers). The method builds on the assumption that not only the the last classifier specializes in the in-domain distribution, last few layers in the backbone beyond the final classifier also focus on the in-domain distribution. So it propose to place the probing classifier at an intermediate layer to improve transferability and adapt to OOD settings

**Strengths:**

1. The paper includes comprehensive experiments demonstrating that intermediate layer probing consistently outperforms last-layer probing in OOD scenarios. This is shown to hold across various network architectures and datasets.

2. By investigating intermediate layer probing, the paper provides a theoretical perspective on measuring distribution shifts within network representations

**Weaknesses:**

1. The definition of “few-shot” may differ from conventional standards, as it uses a significant portion of the test dataset (e.g., half of the CMNIST test samples) rather than the typically small sample sizes seen in few-shot learning. This discrepancy may limit the generalizability of the results.

2. I'm confused that why the zero-shot probing setting can serve as a method to improve the OOD performance. I thought that when trained on in-domain data alone, the OOD performance would not improve since no OOD data is used to train the probing classifier. Would the zero-shot setting can improve the OOD performance?

3. The use of a single-layer linear classifier at an intermediate layer may restrict its ability to capture more complex feature relationships necessary for challenging OOD tasks. Incorporating a more complex classifier, such as a multi-layer perceptron (MLP) or adaptors, could potentially improve performance on these tasks.

4. The main assumption aligns closely with [1], which suggests that layer-wise neural collapse (NC) progresses through two distinct phases in downstream tasks: Phase 1, where representations exhibit a progressively decreasing trend, capturing universal feature mappings; and Phase 2, characterized by a progressively increasing trend, focusing on task-specific feature mappings. A comparison with this prior work would provide valuable insights into how the proposed method builds upon or diverges from these observed phases.

[1] Li, X., Liu, S., Zhou, J., Lu, X., Fernandez-Granda, C., Zhu, Z., & Qu, Q. (2022). Principled and efficient transfer learning of deep models via neural collapse. arXiv preprint arXiv:2212.12206.

**Questions:**

Based on the weakness I have the following questions:

1. Could you clarify how "few-shot" is defined in this context? The use of half the CMNIST test dataset may diverge from traditional definitions of few-shot, which usually refer to very limited samples (e.g., 1–5).

2. Could you provide more details on why zero-shot probing—where only in-domain data is used—seems to enhance OOD performance? This mechanism isn’t entirely clear, as one would generally expect some OOD data to be necessary for adapting to distribution shifts.

3. Has the team considered testing more complex classifiers at the intermediate layer, such as a two-layer MLP? Or add an adaptor between backbone and the linear probing?  Since pruning the intermediate layers may omit useful deep features, a more complex classifier could potentially bridge this gap and enhance OOD results.

**Details Of Ethics Concerns:**

No Ethics Concerns

---

> ### Author Response · Authors · 2024-11-20
>
> Thank you for your insightful comments.
>
> **W1/Q1: Definition of “few-shot” may differ from conventional standards**
>
> We use the term "few-shot" to describe experimental setups where some OOD points are available at training time for probing. While Section 4.2.1 uses a significant portion of the test dataset, Section 4.2.2 explores settings with varying numbers of OOD points, including extremely low availability (as low as 1% of the test set). This corresponds to the following sample sizes per class:
>
> - CIFAR-100C: 2 samples
> - CIFAR-10C: 5 samples
> - CMNIST: 25 samples
> - Waterbirds, CelebA, MultiCelebA: 10 samples
>
> We believe that sample sizes of this magnitude are consistent with the few-shot setting, as they represent minimal data availability while still enabling meaningful evaluation.
>
> **W2/Q2: Reason for ID-trained probes improving OOD generalization isn't clear**
>
> We agree that the result may seem counterintuitive. In Section 5.2 (*Feature Sensitivity in Intermediate and Penultimate Layers*), we define a global metric to measure how sensitive each layer is to distribution shifts. The analysis shows that intermediate layers are consistently less sensitive to such shifts compared to the penultimate layer.
>
> With this setup in mind, our intuition is as follows: intermediate layers capture more general and robust features than the penultimate layer. In contrast, the penultimate layer is optimized for in-distribution performance and is therefore more sensitive to shifts. For example, if an image is perturbed by noise, the features in the penultimate layer may change significantly, while those in intermediate layers remain more stable. Classifiers built on these robust intermediate-layer features are better suited for generalization, even in the zero-shot setting. This is because the OOD representations lie closer to those of ID, enabling the classifier to correctly generalize to OOD points.
>
> **W3/Q3 The use of a single-layer linear classifier at an intermediate layer may restrict its ability to capture more complex feature relationships**
>
> Thank you for pointing this out. While our goal was to present a simple method and challenge previous works claiming the transferability of penultimate-layer features, we agree that incorporating a more complex classifier is an important consideration.
>
> To address this, we conducted additional experiments on CIFAR-10C and CIFAR-100C using ResNet18 and ViT models with an MLP to assess both penultimate and intermediate layer representations. The results, detailed in Appendix C.2 of the updated PDF, showed negligible performance improvements over linear probes for the penultimate layer representations but some improvement for intermediate layer representations. This suggests that while intermediate layers benefit from more complex probes, the limited performance of penultimate layers is not due to the probe’s capacity but rather the absence of sufficient learnable information in the representations.
>
> **W4: Relation to previous work in Neural Collapse**
>
> Thank you for this excellent reference. We believe [1] complements our work, with key differences as follows:
>
> 1. [1] focuses on transfer to new datasets (e.g., from ImageNet-1k pre-trained models), whereas our work examines distribution shifts within the same dataset, including visual shifts (ImageNet, CIFAR) and subpopulation or conditional shifts.
> 2. [1] proposes fine-tuning the layer with the greatest neural collapse (NC$_1$) on the downstream dataset alongside a linear classifier on that layer’s features, whereas our approach keeps the model parameters fixed and does not involve any parameter updates.
>
> To further explore the role of Neural Collapse in dataset-specific models, we followed the setup of [1] and measured NC$_1$ under CIFAR-10C and CIFAR-100C distribution shifts. These results, detailed in Appendix C.1 of the updated PDF, reveal that the penultimate layer consistently exhibits the highest NC$_1$, even under distribution shifts. This contrasts with the findings in [1], which report dataset-dependent variability in the most-collapsed layer. We hypothesize this difference arises because [1] studies general pre-trained models, while our experiments focus on dataset-specific models.

---

> > ### Comment · Reviewer_JuAB · 2024-11-25
> >
> > Thank you for the additional experiments. As most of the concerns are addressed, I would raise the score from 5 to 6.

---

> ### Author Response · Authors · 2024-11-24
>
> Since the discussion phase is closing soon, we wanted to check if our response has addressed your concerns. We’d be happy to hear your thoughts and engage in further discussion if needed.

---

### Author Response · Authors · 2024-12-03

Dear Reviewers,

Thank you for your thoughtful feedback and recognition of our work. We are particularly encouraged by your comments, including:

1. Challenging "the conventional wisdom of using the last layer for zero- and few-shot learning" (`TgHf`).
2. Describing it as a "very intriguing paper" and raising questions about "what happens in the last layers that overcomes generalization to OOD samples" (`ncZg`).
3. Highlighting the "practical benefits, especially for applications where OOD data is scarce or unavailable" (`G9DE`).
4. Recognizing that the paper is "well-written and easy to follow" (`Nh8y`, `G9DE`, `TgHf`, `P4oy`).
5. Acknowledging the clarity and thoroughness of our experiments, which evaluate intermediate-layer performance across various datasets and architectures (`G9DE`, `JuAB`).

We also greatly appreciate your valuable suggestions, which led to updates including:

- A PCA-based analysis confirming that the improvement in ILCs' performance is not due to feature dimensionality (`P4oy`).
- Experiments with non-linear probes, showing marginal improvements in the penultimate layer and supporting the robustness of linear probes (`JuAB`).
- An analysis of Neural Collapse across layers, highlighting differences in robustness (`JuAB`).
- Results validating the transferability of findings across datasets (`TgHf`).
- Extensions to related work, specifying how our work relates to prior research (`G9DE`, `JuAB`).

We also value the exciting questions and suggestions raised by reviewers for potential future work. Your feedback has strengthened the work, and we are grateful for your engagement!

---

### Meta-Review · Area_Chair_nHnS · 2024-12-18

**Metareview:**

(a) summary

This paper investigates whether the intermediate layers of a deep neural network offer better out-of-distribution (OOD) generalization than earlier and last layers. It demonstrates that in a number of settings, including zero- and few-shot, training classifiers on intermediate layer representations tends to perform better than last-layer retraining, suggesting intermediate layers are less sensitive to distribution shifts.

(b) strengths
+ The paper is well motivated: it explores what layers are robust to distribution shift which is important for OOD generalization.
+ It introduces a new metric "sensitivity score" for measuring layer sensitivity to distribution shift, feature sensitivity.
+ The interesting observation challenges the common practice of relying solely on the last layer's representations.
+ It is well-written and easy to follow.

(c) weaknesses
- It has limited novelty: the novelty of this contribution is somewhat diminished given that an earlier paper (accepted at last year's ICLR) has also proposed evaluating intermediate layers’ effectiveness in generalization for OOD samples:

[2] Gerritz et al. (2024) "Zero-shot generalization across architectures for visual classification." The Second Tiny Papers Track at ICLR 2024, https://openreview.net/forum?id=orYMrUv7eu

An extended version of that paper (from the same group) also used linear probes to quantify the out-of-sample accuracy of intermediate representations:

[3] Dyballa et al. (2024). A separability-based approach to quantifying generalization: which layer is best?. arXiv preprint arXiv:2405.01524, https://arxiv.org/abs/2405.01524

- The experiments do not include some datasets and models.
- The results are not significant for transformer but CNN.
- It lacks details on some concepts such as "information content".
- It is not practical because it is not easy to find the best layer.

(d) decision

This paper is well motivated: it explores what layers are robust to distribution shift which is important for OOD generalization. The interesting observation challenges the common practice of relying solely on the last layer's representations. The authors have a successful rebuttal, and all reviewers recommend accept.

**Additional Comments On Reviewer Discussion:**

The main concerns of the reviewer are on the limited novelty and missing experiments. The authors' rebuttal and added experiments helped to address the concerns by the reviewers. All reviewers raised their scores to accept.

---

### Decision · Program_Chairs · 2025-01-22

Accept (Poster)